# LIFT YOUR MOLECULES: MOLECULAR GRAPH GENERATION IN LATENT EUCLIDEAN SPACE

**Mohamed Amine Ketata**[*], **Nicholas Gao**[*], **Johanna Sommer**[*],
**Tom Wollschläger, Stephan Günnemann**
Department of Computer Science & Munich Data Science Institute
Technical University of Munich
`{a.ketata,n.gao,jm.sommer,t.wollschlaeger,s.guennemann}@tum.de`

## ABSTRACT

We introduce a new framework for 2D molecular graph generation using 3D molecule generative models. Our Synthetic Coordinate Embedding (SYCO) framework maps 2D molecular graphs to 3D Euclidean point clouds via synthetic coordinates and learns the inverse map using an E($n$)-Equivariant Graph Neural Network (EGNN). The induced point cloud-structured latent space is well-suited to apply existing 3D molecule generative models. This approach simplifies the graph generation problem into a point cloud generation problem followed by node and edge classification tasks, without relying on molecular fragments nor autoregressive decoding. Further, we propose a novel similarity-constrained optimization scheme for 3D diffusion models based on inpainting and guidance. As a concrete implementation of our framework, we develop EDM-SYCO based on the E(3) Equivariant Diffusion Model (EDM). EDM-SYCO achieves state-of-the-art performance in distribution learning of molecular graphs, outperforming the best non-autoregressive methods by more than 26% on ZINC250K and 16% on the GuacaMol dataset while improving conditional generation by up to 3.9 times.[1]

## 1 INTRODUCTION

Machine learning-based drug discovery has recently shown great potential to accelerate the drug development process while reducing costs associated with clinical trials (Jiménez-Luna et al., 2020; Kim et al., 2020). Among the steps of this process, generating novel molecules or optimizing properties such as drug-likeness or synthesizability are of crucial importance. Molecular graph generation is an active area of research that aims to solve these tasks (Gómez-Bombarelli et al., 2018).

Molecular graph generation is challenging due to the discrete and sparse nature of graphs as well as the presence of symmetric substructures such as rings. Many methods have been proposed to tackle this problem, offering different tradeoffs (Jin et al., 2018; Shi et al., 2020). A common approach is to train a Variational Autoencoder (VAE) to encode the graph into a continuous fixed-size latent space, then decode it back – oftentimes autoregressively – node-by-node (Liu et al., 2018) or fragment-by-fragment (Jin et al., 2018; 2020; Maziarz et al., 2021). While such methods achieve good generation and optimization performance, they have inherent limitations. First, using a fixed-sized latent space does not account for the variable size of molecular graphs, which can go from a few atoms for small molecules to thousands for macromolecules. Second, autoregressive decoding requires choosing a concrete generation order (Schneider et al., 2015), which prevents learning permutation-invariant graph distributions. Fueled by recent developments in diffusion models (Ho et al., 2020; Austin et al., 2021), new approaches to molecular graph generation have emerged that overcome these limitations (Jo et al., 2022; Vignac et al., 2022; Jo et al., 2023). These methods define a discrete diffusion process directly on the node and edge attributes and learn to reverse it to generate graphs all at once, i.e., not autoregressively. However, they still lack behind autoregressive methods, struggling to accurately estimate the joint distribution of nodes and edges.

---

[*]Equal contribution.
[1]Our code is available at https://github.com/ketatam/SyCo

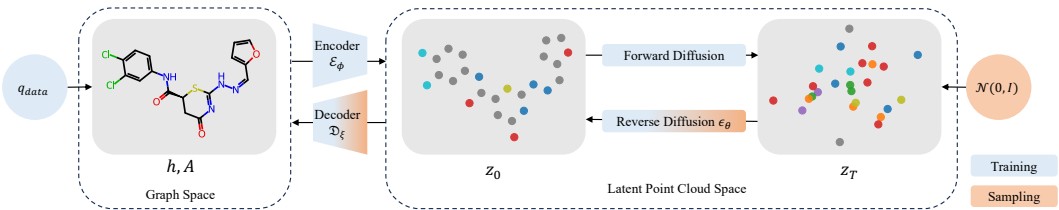

Figure 1: Overview of EDM-SyCo. *(Training)* First, the autoencoder is trained to map between molecular graphs and latent Euclidean point clouds. Then, the diffusion model is trained on the fixed latent space. *(Sampling)* Starting with a Gaussian sample, the diffusion model denoises it for $T$ steps to predict the clean point cloud, which is mapped to a molecular graph using the decoder.

In this work, we develop a method that combines the benefits of all-at-once generation with the benefits of a continuous latent space while avoiding the information bottleneck of fixed-sized latent spaces. Inspired by recent developments in the adjacent field of 3D molecule generation, we propose to embed molecular graphs as latent 3D point clouds that implicitly encode the discrete graph structure in *synthetic* Euclidean coordinates and use point cloud generative models to learn their distribution. We call this framework Synthetic Coordinate Embedding (SyCo).

When combined with EDM (Hoogeboom et al., 2022), our framework gives rise to an atom-based, all-at-once, and permutation-invariant generative model for molecular graphs, which we call EDM-SyCo, depicted in Figure 1. On the ZINC250K dataset, EDM-SyCo outperforms existing all-at-once diffusion-based models by more than 26% on distribution learning benchmarks and by up to 15.6 times on conditional generation benchmarks, closing the gap to autoregressive models and highlighting the benefit of our continuous latent space on generation and optimization. In addition, we propose a new similarity-constrained optimization procedure for diffusion models on point clouds without retraining or specialized architectures, visualized in Figure 3. Notably, our proposed algorithm enables the addition of new atoms during the reverse diffusion process, addressing a well-known limitation of prior 3D diffusion models, namely the fixed number of nodes.

## 2 RELATED WORK

**Molecular Graph Generation**   Early work on molecule generation (Gómez-Bombarelli et al., 2018; Segler et al., 2018) used language models to generate SMILES (Weininger, 1988), a text-based representation of molecules. However, this type of representation cannot capture the inherent structure of molecules well, which are more accurately depicted as graphs. For instance, small changes in the graph structure may correspond to large differences in the SMILES string (Jin et al., 2018). Therefore, recent methods have focused on graph-based approaches that can be broadly classified into two categories. All-at-once generation methods generate nodes and edges in parallel, often in a multi-step process such as diffusion models (Niu et al., 2020; Jo et al., 2022; Vignac et al., 2022). Autoregressive methods specify a generation order and generate molecules either atom-by-atom (Li et al., 2018; Shi et al., 2020) or fragment-by-fragment based on a pre-computed fragment vocabulary (Jin et al., 2018; 2020; Maziarz et al., 2021; Kong et al., 2022; Geng et al., 2023). EDM-SyCo is an all-at-once molecular graph generative model that operates on the atom level.

**Molecule Generation in 3D**   Another line of work aims to generate molecules in 3D Euclidean space and solve a point cloud generation problem (Gebauer et al., 2019; Garcia Satorras et al., 2021; Ayadi et al., 2025). For instance, EDM (Hoogeboom et al., 2022) and GeoLDM (Xu et al., 2023) are diffusion-based models for 3D molecule generation that jointly generate atomic features and coordinates. While tangentially related, these methods solve a different problem from molecular graph generation and require molecules with 3D information for training. Our proposed SyCo framework is an attempt to connect these two lines of work by enabling the training of 3D generative models on graph datasets through synthetic coordinates, formally introducing new graph generation methods. While we are the first to explore synthetic coordinates for generative models, Gasteiger et al. (2021) demonstrated that synthetic coordinates improve molecular property prediction tasks.

**Latent Generative Models**   The idea of defining the generative model on a space different from the original data space is reminiscent of latent generative models (Dai & Wipf, 2019; Vahdat et al.,

2021). Recently, this approach has become increasingly popular with latent diffusion models. Stable Diffusion models (Rombach et al., 2022) achieve impressive results on text-guided image generation, and GeoLDM (Xu et al., 2023) extends this to molecular point cloud generative models. While EDM-SYCO operates on a similar latent space to GeoLDM, their fundamental difference lies in the original data space. Our model maps discrete molecular graphs to continuous point clouds and can be seen as a cross-modality latent generative model. In contrast, GeoLDM embeds point clouds as new point clouds with a reduced feature dimension.

## 3 SYNTHETIC COORDINATE EMBEDDING (SYCO) FRAMEWORK

Our goal is to model molecular graph distributions through distributions of Euclidean point clouds that implicitly encode the graph structure into their coordinates. For this, we propose a novel autoencoder architecture that maps between molecular graphs and 3D point cloud representations. By training it using a reconstruction objective, we can reformulate the problem of generating discrete molecular graphs to an equivalent problem of generating 3D point clouds defined on the induced latent space.

**Note on equivariance** When dealing with physical objects like molecules, it is important to consider *equivariance*. Formally, a function $f : A \to B$ is *equivariant* to the action of a group $G$ iff $f(T_g^A(z)) = T_g^B(f(z))$ for all $g \in G$, where $T_g^A$ and $T_g^B$ are the representations of the group element $g$ in $A$ and $B$ respectively (Serre et al., 1977). As a special case, it is *invariant* iff $f(T_g^A(z)) = f(z)$. In the context of generative modeling, we aim to learn distributions invariant to the symmetries of the data, namely graph permutations in our case. In practice, this has the benefit of not relying on any generation order and enabling efficient likelihood estimation since an invariant likelihood is identical for all permutations of the same graph. We leverage equivariant architectures such as EGNNs to learn such distributions, removing the need for data augmentation during training, further increasing efficiency. EGNNs (Satorras et al., 2021) are a special type of graph neural networks that learn functions equivariant to rotations, translations, reflections, and permutations of their input point cloud. More details on this architecture are given in Appendix D. Also note that due to the reflection equivariance, our model is invariant to chirality.

**Notation** We introduce helpful notation for the two molecule representations used in this work. Let $N$ denote the number of atoms of a given molecule. On the one hand, the molecular graph $\mathcal{G} = (h, A) \in \mathbb{G}$ consists of atoms as nodes and chemical bonds as edges. Each atom has one of $a$ atom types and one of $c$ formal charges, while each bond has one of $b$ bond types. We represent atoms as stacked one-hot encodings $h \in \{0, 1\}^{N \times (a+c)}$ and chemical bonds as one-hot vectors in an adjacency matrix $A \in \{0, 1\}^{N \times N \times b}$. Note that the molecular graph is sometimes called a "2D molecule" as it can be typically drawn as a planar graph. On the other hand, the molecular point cloud $\mathcal{P} = (h, x) \in \mathbb{P}$ is a 3D point cloud, where $x \in \mathbb{R}^{N \times 3}$ represents the coordinates of atoms' nuclei in Euclidean space. The point cloud encodes the bond information implicitly in the atomic coordinates. We now present our autoencoder model, whose overview is shown in Figure 2.

**Encoding** The encoder $\mathcal{E}_\phi : \mathbb{G} \to \mathbb{R}^{N \times (d+3)}$ with parameters $\phi$ maps a molecule's graph representation $\mathcal{G} = (h, A)$ to its point cloud representation $\mathcal{P} = (h, x)$, then further compresses it into a node-structured latent representation $z = (z^{(h)}, z^{(x)})$, where $z^{(h)} \in \mathbb{R}^{N \times d}$ and $z^{(x)} \in \mathbb{R}^{N \times 3}$ are continuous embeddings for $h$ and $x$ with $d < a + c$ and $z \in \mathbb{R}^{N \times (d+3)}$ is their concatenation.

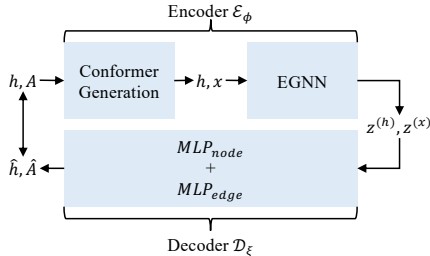

In the first encoding step, the coordinates $x$ are computed using the ETKDG algorithm (Riniker & Landrum, 2015), which first computes a molecule's distance bounds matrix based on typical bond lengths between different atom types, then produces atomic coordinates that satisfy these bounds, which are further optimized using a molecular force field. Crucially, this procedure allows to leverage chemical domain knowledge as an inductive bias into the

Figure 2: Autoencoder architecture. The encoder maps a molecular graph to a point cloud, and the decoder learns the inverse. Both are trained jointly to minimize the reconstruction loss.

model and get chemically meaningful 3D conformers. These coordinates encode the bond information of the molecular graph and ideally define an injective mapping from $\mathbb{G}$ to $\mathbb{P}$. Note that this step is not learned and, thus, not required to be differentiable. In practice, we generate conformers for the whole training dataset in a preprocessing step and reuse them throughout training.

Then, we apply an EGNN that maps the intermediate representation $\mathcal{P} = (\boldsymbol{h}, \boldsymbol{x})$ to the latent representation $\boldsymbol{z} = (\boldsymbol{z}^{(h)}, \boldsymbol{z}^{(x)})$, defining the space of the generative model. The main benefit of this is to incorporate the neighborhood information into each atom's embeddings and coordinates and mapping the discrete features $\boldsymbol{h}$ into a continuous and lower-dimensional embedding $\boldsymbol{z}^{(h)}$.

**Decoding** The decoder $\mathcal{D}_\xi : \mathbb{R}^{N \times (d+3)} \to \mathbb{G}$ with parameters $\xi$ aims to approximate the inverse function $\mathcal{E}_\phi^{-1}$ and maps the latent point cloud $\boldsymbol{z}$ of a molecule back to its graph representation $\mathcal{G}$. It consists of an atom and a bond classifier. For each node $i$, the decoder applies a 2-layer Multi-Layer Perceptron (MLP) on its latent embedding to predict the logits for the atom type and the formal charge as $\hat{h}_i = \text{MLP}_{node}(z_i^{(h)})$. For each pair of nodes $i$ and $j$, the bond type is inferred by running a different 2-layer MLP on their edge features to obtain the corresponding logits as follows:

$$\hat{A}_{ij} = \frac{1}{2} \left( \text{MLP}_{edge} \left( \begin{bmatrix} z_i^{(h)} \\ z_j^{(h)} \\ d_{ij}^2 \end{bmatrix} \right) + \text{MLP}_{edge} \left( \begin{bmatrix} z_j^{(h)} \\ z_i^{(h)} \\ d_{ij}^2 \end{bmatrix} \right) \right),$$

where $d_{ij} = \|z_i^{(x)} - z_j^{(x)}\|_2$ and we use the simple averaging trick to ensure that $\hat{\boldsymbol{A}}$ is symmetric.

**Autoencoder Training** The encoder and decoder are jointly optimized in a variational autoencoder (VAE) framework. We define probabilistic encoding and decoding processes as $q_\phi(\boldsymbol{z}|\mathcal{G}) = \mathcal{N}(\mathcal{E}_\phi(\mathcal{G}), \sigma_0^2 \boldsymbol{I})$ and $p_\xi(\mathcal{G}|\boldsymbol{z}) = \prod_{i=1}^N p_\xi(h_i|\boldsymbol{z}) \prod_{j=1}^N p_\xi(A_{ij}|\boldsymbol{z})$, respectively, where $\sigma_0 \in \mathbb{R}$ controls the variance in the latent space. The VAE is trained by minimizing the loss function

$$\mathcal{L}_{VAE} = -\mathbb{E}_{q(\mathcal{G})q_\phi(\boldsymbol{z}|\mathcal{G})} \left[ p_\xi(\mathcal{G}|\boldsymbol{z}) \right], \tag{1}$$

which, in practice, is computed as the cross-entropy loss. The decoder performs three classification tasks to reconstruct a molecular graph: atom types, formal charges, and bond types. The total loss is the unweighted sum of the corresponding three cross-entropy terms.

Before continuing with our approach, we present an ablation study on our autoencoder architecture. The task consists of predicting all the bond types of a molecule from its 3D coordinates. We compare our EGNN-based approach with (i) an MLP and (ii) a rule-based system akin to Hoogeboom et al. (2022). More details on the experimental setup are in Appendix G.1. Results on ZINC250K are shown in Table 1, and similar results are observed for GuacaMol. Our autoencoder model achieves an almost perfect reconstruction accuracy, highlighting the importance of the EGNN in our architecture. This experiment supports our assumptions (i) that the used mapping from $\mathbb{G}$ to $\mathbb{P}$ is injective and (ii) that the decoder can accurately approximate its inverse.

Table 1: Accuracy of predicting bond types from 3D coordinates on ZINC250K. Existing methods use rule-based algorithms, while we introduce an EGNN-based algorithm.

| METHOD | ACCURACY |
|---|---|
| EGNN + MLP | **99.93%** |
| MLP | 11.77% |
| RULE-BASED | 11.78% |

## 4 EDM-SYCO

Our SYCO framework lifts the graph generation problem to a point cloud generation problem in 3D. Concretely, we are interested in learning the distribution of latent point clouds $\boldsymbol{z} \sim q_\phi(\boldsymbol{z}|\mathcal{G})q(\mathcal{G})$ defined by the underlying molecular graph distribution and the trained encoder by approximating it with a parametric distribution $p_\theta$. While this generation problem can, in principle, be approached by *any* 3D molecular generative method, this work adapts EDM under the SYCO framework due to its simplicity and empirical performance in 3D molecule generation tasks (Hoogeboom et al., 2022) .

### 4.1 EQUIVARIANT DIFFUSION ON LATENT POINT CLOUDS

A diffusion model generates samples by reversing a diffusion process, the process of adding noise to data (Sohl-Dickstein et al., 2015; Ho et al., 2020). For ease of notation, we flatten the latent repre-

sentation of the point clouds $\boldsymbol{z}$, namely their latent coordinates and features, into a one-dimensional vector denoted as $\boldsymbol{z}_0 \in \mathbb{R}^{d'}$ with $d' = N(d+3)$. The *diffusion* or *forward process* is a fixed Markov chain of Gaussian updates that transform $\boldsymbol{z}_0$ into latent variables $\boldsymbol{z}_1, \ldots, \boldsymbol{z}_T$ of the same dimension as $\boldsymbol{z}_0$, defined as

$$q(\boldsymbol{z}_{1:T}|\boldsymbol{z}_0) := \prod_{t=1}^{T} q(\boldsymbol{z}_t|\boldsymbol{z}_{t-1}), \text{ with } q(\boldsymbol{z}_t|\boldsymbol{z}_{t-1}) := \mathcal{N}(\boldsymbol{z}_t; \sqrt{1-\beta_t}\boldsymbol{z}_{t-1}, \beta_t \boldsymbol{I}), \tag{2}$$

where $\beta_t$ is typically chosen such that $q(\boldsymbol{z}_T)$ converges to $\mathcal{N}(\boldsymbol{z}_T; \boldsymbol{0}, \boldsymbol{I})$. By defining $\alpha_t := 1 - \beta_t$ and $\bar{\alpha}_t := \prod_{i=1}^{t} \alpha_i$, we can also map $\boldsymbol{z}_0$ directly to $\boldsymbol{z}_t$ through

$$q(\boldsymbol{z}_t|\boldsymbol{z}_0) = \mathcal{N}(\boldsymbol{z}_t; \sqrt{\bar{\alpha}_t}\boldsymbol{z}_0, (1-\bar{\alpha}_t)\boldsymbol{I}). \tag{3}$$

To reverse the above process, we start from a Gaussian noise sample $\boldsymbol{z}_T$ and iteratively sample from the posterior distributions $q(\boldsymbol{z}_{t-1}|\boldsymbol{z}_t)$, which is also Gaussian if $\beta_t$ is small (Sohl-Dickstein et al., 2015). However, estimating them requires using the entire dataset. Therefore, diffusion models learn a *reverse process* $p_\theta(\boldsymbol{z}_{0:T}) := p(\boldsymbol{z}_T) \prod_{t=1}^{T} p_\theta(\boldsymbol{z}_{t-1}|\boldsymbol{z}_t)$, with $p(\boldsymbol{z}_T) = \mathcal{N}(\boldsymbol{z}_T; \boldsymbol{0}, \boldsymbol{I})$ and

$$p_\theta(\boldsymbol{z}_{t-1}|\boldsymbol{z}_t) := \mathcal{N}(\boldsymbol{z}_{t-1}; \boldsymbol{\mu}_\theta(\boldsymbol{z}_t, t), \sigma_t^2 \boldsymbol{I}), \tag{4}$$

where the mean $\boldsymbol{\mu}_\theta(\boldsymbol{z}_t, t) \in \mathbb{R}^{d'}$ is parametrized by a neural network.

We follow Ho et al. (2020) and, instead of directly predicting the mean, predict the Gaussian noise added to the latent point cloud representation during the diffusion process. The mean then becomes $\boldsymbol{\mu}_\theta(\boldsymbol{z}_t, t) = \frac{1}{\sqrt{\alpha_t}}\left(\boldsymbol{z}_t - \frac{1-\alpha_t}{\sqrt{1-\bar{\alpha}_t}}\boldsymbol{\epsilon}_\theta(\boldsymbol{z}_t, t)\right)$, where $\boldsymbol{\epsilon}_\theta(\boldsymbol{z}_t, t) \in \mathbb{R}^{d'}$ is the output of the neural network. Following Hoogeboom et al. (2022), we parametrize $\boldsymbol{\epsilon}_\theta$ via an EGNN (Satorras et al., 2021). Additionally, the reverse process ensures equivariance to translations by defining the learned distribution $p_\theta(\boldsymbol{z}_0)$ on the zero center of gravity subspace.

This network can then be trained by optimizing the $L_2$ loss between the sampled Gaussian noise at each step and the network's output,

$$\mathcal{L}_{DM} = \mathbb{E}_{\boldsymbol{z}_0, \boldsymbol{\epsilon}, t}\left[\|\boldsymbol{\epsilon} - \boldsymbol{\epsilon}_\theta(\boldsymbol{z}_t, t)\|_2^2\right]. \tag{5}$$

During sampling, we start from a Gaussian sample $\boldsymbol{z}_T \sim p(\boldsymbol{z}_T)$ and iteratively denoise it by sampling $\boldsymbol{\epsilon}_t \sim \mathcal{N}(\boldsymbol{0}, \boldsymbol{I})$ and transforming $z_t$ to $z_{t-1}$:

$$\boldsymbol{z}_{t-1} = \underbrace{\frac{1}{\sqrt{\alpha_t}}(\boldsymbol{z}_t - \frac{1-\alpha_t}{\sqrt{1-\bar{\alpha}_t}}\boldsymbol{\epsilon}_\theta(\boldsymbol{z}_t, t))}_{\boldsymbol{\mu}_\theta(\boldsymbol{z}_t, t)} + \underbrace{\frac{\sqrt{1-\bar{\alpha}_{t-1}}}{\sqrt{1-\bar{\alpha}_t}}\sqrt{\beta_t}\,\boldsymbol{\epsilon}_t}_{\sigma_t}. \tag{6}$$

## 4.2 Training and Sampling from EDM-SyCo

After training the autoencoder as described in Section 3, EDM is trained in a second stage by optimizing the loss function defined in Equation 5. We describe the full training procedure of both parts in Algorithm 1. Consistent with previous work on latent diffusion models (Rombach et al., 2022; Xu et al., 2023), we found this two-stage training procedure to perform better than jointly training the autoencoder and the diffusion model.

For sampling, we need access to the trained EDM and the decoder but not the encoder. First, starting with a sample $\boldsymbol{z}_T \sim \mathcal{N}(\boldsymbol{0}, \boldsymbol{I})$ representing a random 3D point cloud, we iteratively denoise it with the diffusion model (Equation 6) to obtain the clean sample $\boldsymbol{z}_0$. Then, we apply the decoder to get the corresponding molecular graph $\boldsymbol{h}, \boldsymbol{A} = \mathcal{D}_\xi(\boldsymbol{z}_0)$. Algorithm 2 formally describes this procedure, and Figure 1 further illustrates it. This sampling procedure defines a permutation-invariant molecular graph distribution. The following proposition, which we prove in Appendix C.2, formalizes this:

**Proposition 4.1.** *The marginal distribution of molecular graphs $p_{\theta,\xi}(\mathcal{G}) = \mathbb{E}_{p_\theta(\boldsymbol{z}_0)}[p_\xi(\mathcal{G}|\boldsymbol{z}_0)]$ defined by the EDM and the decoder, is an $S_N$-invariant distribution, i.e. for any molecular graph $\mathcal{G} = (\boldsymbol{h}, \boldsymbol{A})$, $p_{\theta,\xi}(\boldsymbol{h}, \boldsymbol{A}) = p_{\theta,\xi}(P\boldsymbol{h}, PAP)$ for any permutation matrix $P \in S_N$.*

So far, we have assumed that the number of atoms $N$ is fixed across all molecules. In practice, to generate molecules with different sizes, we follow previous work (Hoogeboom et al., 2022) and estimate the empirical distribution of the number of atoms in the training set $p(N)$ and use it to sample different numbers of atoms before running the sampling procedure described above. In Appendix F.2, we describe the use of conditional distributions to sample $N$ in a constrained generation or optimization.

## 5 CONTROLLABLE GENERATION AND SIMILARITY-CONSTRAINED OPTIMIZATION

So far, we have introduced our approach to distribution learning on molecular graphs. However, many tasks in drug discovery require generating molecules with specific conditions such as chemical properties, similarity constraints, or the presence of certain desirable substructures. In this section, we describe how we can adapt the sampling procedure of EDM-SYCO (or other models combined with SYCO) – trained in an unconditional setting – to different conditioning modalities. While previous work on 3D molecule generation tackled conditional generation (Hoogeboom et al., 2022; Bao et al., 2022) and scaffolding (Schneuing et al., 2022), the similarity-constrained optimization task (Jin et al., 2020) has not been addressed in the 3D molecule generation literature. We fill this gap by introducing a novel sampling algorithm for diffusion models to perform similarity-constrained optimization.

**Property-Conditional Generation via Diffusion Guidance**    In a conditional generation setting, we want to generate a molecule with a specific property $c$ by sampling from the conditional distribution $q(\boldsymbol{z}|c)$. To achieve this, we adapt the classifier guidance algorithm (Dhariwal & Nichol, 2021), initially proposed to guide image diffusion models using a classifier. Instead, we use a property regressor to guide the generation towards molecules with specific continuous properties similar to Bao et al. (2022). One can show that the denoising network's output relates to the score function of the data distribution via $\nabla_{\boldsymbol{z}_t} \log q(\boldsymbol{z}_t) \approx -\frac{1}{\sqrt{1-\bar{\alpha}_t}}\boldsymbol{\epsilon}_\theta(\boldsymbol{z}_t, t)$ (Dhariwal & Nichol, 2021) and the sampling procedure of diffusion models (Equation 6) is equivalent to running Langevin dynamics using the score function $\nabla_{\boldsymbol{z}_t} \log q(\boldsymbol{z}_t)$ (Welling & Teh, 2011; Song & Ermon, 2019). This provides a way to sample from the distribution $q$ only through its score function. For the case of $q(\boldsymbol{z}_t|c)$, its score function can be written using Bayes' rule and approximated as:

$$\nabla_{\boldsymbol{z}_t} \log q(\boldsymbol{z}_t|c) = \nabla_{\boldsymbol{z}_t} \log q(\boldsymbol{z}_t) + \nabla_{\boldsymbol{z}_t} \log q(c|\boldsymbol{z}_t)$$
$$\approx -\frac{1}{\sqrt{1-\bar{\alpha}_t}}(\boldsymbol{\epsilon}_\theta(\boldsymbol{z}_t, t) - \sqrt{1-\bar{\alpha}_t}\nabla_{\boldsymbol{z}_t} \log p_\eta(c|\boldsymbol{z}_t)),$$

where $p_\eta(c|\boldsymbol{z}_t) \sim \exp(-s(g_\eta(\boldsymbol{z}_t) - c)^2)$ is a Gaussian distribution approximating the probability of molecule $\boldsymbol{z}_t$ having property $c$, $g_\eta$ is a property regressor trained to predict the property $c$ given a noisy molecule $\boldsymbol{z}_t$, and $s$ is a scale factor that controls the skewness of the distribution. With this, we can rewrite the conditional score function as $\nabla_{\boldsymbol{z}_t} \log q(\boldsymbol{z}_t|c) \approx -\frac{1}{\sqrt{1-\bar{\alpha}_t}}(\boldsymbol{\epsilon}_\theta(\boldsymbol{z}_t, t) + \sqrt{1-\bar{\alpha}_t}s\nabla_{\boldsymbol{z}_t}(g_\eta(\boldsymbol{z}_t) - c)^2)$. To sample from this distribution, we use the same sampling algorithm defined by Equation 6 and replace $\boldsymbol{\epsilon}_\theta(\boldsymbol{z}_t, t)$ by

$$\boldsymbol{\epsilon}_\theta(\boldsymbol{z}_t, t) + \sqrt{1-\bar{\alpha}_t}s\nabla_{\boldsymbol{z}_t}(g_\eta(\boldsymbol{z}_t) - c)^2. \tag{7}$$

Intuitively, this can be seen as minimizing the loss $(g_\eta(\boldsymbol{z}_t) - c)^2$. Appendix E.2 provides additional details and discussions about the equivariance of the regressor.

**Unconstrained Property Optimization via Diffusion Guidance**    With the formulation introduced above, we can sample molecules with high property values by simply using the denoising step

$$\boldsymbol{\epsilon}_\theta(\boldsymbol{z}_t, t) - \sqrt{1-\bar{\alpha}_t}s\nabla_{\boldsymbol{z}_t}g_\eta(\boldsymbol{z}_t). \tag{8}$$

Notice that we replaced the loss with the negative gradient of the regressor.

**Scaffold-Constrained Generation via Inpainting**    For tasks such as structure-based drug design, it is often desirable to generate molecules that contain a specific substructure, usually called a *scaffold* (Schuffenhauer et al., 2007; Maziarz et al., 2021). This problem can be viewed as a completion task and is reminiscent of inpainting, which aims to complete missing parts of an image (Song et al., 2020; Lugmayr et al., 2022). Lugmayr et al., 2022 propose RePaint, a simple modification to the sampling procedure of denoising diffusion models, enabling them to perform inpainting. This approach has already been successfully applied to molecule generation in 3D (Schneuing et al., 2022).

The core idea of RePaint is to start the sampling procedure from a random sample $\boldsymbol{z}_T$ and, at each step, enforce the part that we want to be present in the final sample. To illustrate this for the case of molecules, let $\tilde{\boldsymbol{z}}_0$ denote the point cloud of a molecule containing the desired scaffold. Let $m$ be a binary mask that indicates which nodes from $\tilde{\boldsymbol{z}}_0$ belong to the scaffold such that $m \odot \tilde{\boldsymbol{z}}_0$ denotes the

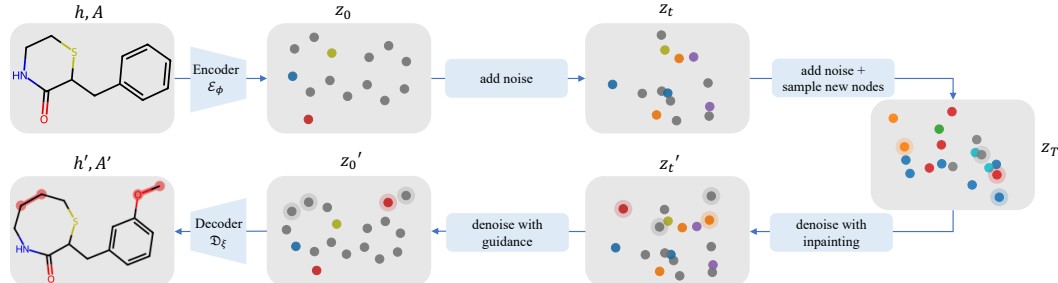

Figure 3: Overview of our constrained optimization procedure. Based on a noising/denoising approach, we run the first steps of the reverse diffusion process using the inpainting algorithm to add new atoms and the remaining steps using the guidance algorithm to increase the target property. The depicted molecules have QED values of 0.79 (initial) and 0.91 (optimized), with a 53% similarity.

scaffold point cloud and $(1 - m) \odot \tilde{z}_0$ the unknown part. We aim to sample a new point cloud $z$ containing this scaffold while extending it to get a harmonized point cloud. Formally, we require that $m \odot z_0 = m \odot \tilde{z}_0$. To achieve this, we iteratively apply the following modified sampling step starting from $t = T$:

$$z_{t-1}^{\text{scaffold}} \sim \mathcal{N}(\sqrt{\bar{\alpha}_{t-1}}\tilde{z}_0, (1 - \bar{\alpha}_{t-1})\boldsymbol{I}) \tag{9}$$

$$z_{t-1}^{\text{unknown}} \sim \mathcal{N}(\boldsymbol{\mu}_\theta(z_t, t), \sigma_t^2 \boldsymbol{I}) \quad \text{(as in Equation 6)} \tag{10}$$

$$z_{t-1} = m \odot z_{t-1}^{\text{scaffold}} + (1 - m) \odot z_{t-1}^{\text{unknown}} \tag{11}$$

We refer to Appendix E.3 for more implementation details.

**Similarity-Constrained Optimization** In this task, we aim to improve the target properties of a given molecule while satisfying a similarity constraint. We propose a novel approach to this task by combining the regressor guidance algorithm and the inpainting algorithm discussed above. The key idea is to add noise to the initial molecule and then denoise it with the regressor guidance algorithm to improve the target property. Formally, starting from the initial point cloud $z_0$ with $N$ atoms, we sample a noised $z_t$ following Equation 3 for $t \in (0, T)$. $t$ is a hyperparameter that controls the optimization quality. On the one hand, increasing $t$ results in more reverse diffusion steps with the guidance algorithm, thus increasing the chance of improving the target property. On the other hand, it also results in losing information about the initial molecule due to noise. Intuitively, the best $t$ yields the highest property improvement while satisfying the similarity constraints.

However, since the diffusion model operates on $N$ points, the denoised molecule also has $N$ atoms. This limits the optimization of a molecule's property as it prevents adding new atoms. To overcome this limitation, we sample a Gaussian point cloud $z_T$ with $N'$ atoms, where $N' > N$, and run the inpainting algorithm from $T$ to $t$ with $z_t$ as the scaffold, meaning that $z_t$ will replace $m \odot \tilde{z}_0$ in the inpainting algorithm described above. This results in a new intermediate representation $z_t'$ whose first $N$ points correspond to $z_t$, and the rest are the newly added atoms, as desired. Finally, $z_t'$ is denoised from $t$ to 0 using the guidance algorithm with maximization objective (Equation 8) to get the point cloud $z_0'$ of the optimized molecule, which is then decoded to the graph using the decoder $\mathcal{D}_\xi$. An overview of this procedure and an example test molecule are shown in Figure 3.

## LIMITATIONS

EDM-SYCO can be seen as an extension to EDM, which carries the limitations of diffusion models like high inference cost. To mitigate this issue in future work, we can incorporate more efficient sampling strategies (Karras et al., 2022; 2023), or use more efficient models such as recent conditional flow matching approaches (Tong et al., 2023; Song et al., 2023). In addition, our sampling approach requires specifying the number of nodes a priori. While this works well empirically, we leave tackling the fundamental issue of generating variable-sized graphs or point clouds to future work. Lastly, our constrained optimization procedure cannot remove atoms during inference. Our experiments found that the decoder tends to predict disconnected subgraphs, which we treat as removing atoms, but a rigorous solution may improve performance.

## 6 EXPERIMENTS

### 6.1 COMPARISON TO 3D MOLECULE GENERATIVE MODEL GEOLDM

EDM-SYCO leverages EDM to learn the distribution of 3D molecules in latent space, and combines it with an autoencoder that maps between 2D and 3D molecules. The overall architecture is therefore similar to GeoLDM in that both are EDM-based latent diffusion models for molecules. The main differences lie, however, in the design of the autoencoder and in the training and sampling procedure. In this section, we explain the need for such changes in order to adapt 3D molecule generative models like GeoLDM to the different modality of 2D molecules and validate the effect of these changes experimentally. In essence, there are two main challenges when it comes to applying GeoLDM or any other 3D molecule generative models to 2D molecule generation:

*(i) GeoLDM requires datasets with 3D atomic coordinates for training.* To enable training on 2D datasets like ZINC250K or GuacaMol, we leverage the conformer generation component introduced in Section 3. Crucially, this allows to inject chemical domain knowledge as an inductive bias into the model by leveraging typical bond lengths and molecular force fields when computing 3D conformers.

*(ii) GeoLDM uses a simple rule-based algorithm to predict 2D molecules from 3D molecules, which fails even on medium-sized molecules.* This algorithm predicts bond types from inter-atomic distances using a lookup table. In our experiments, this algorithm can correctly predict all bond types of a molecule only on 11.78% of the ZINC250K dataset (See Table 1). To overcome this limitation, we introduce our EGNN-based bond type predictor, which can leverage the richer neighborhood information of an atom and does not rely on hard-coded rules. To highlight the effect of this change on the quality of the generated 2D graphs, we compare GeoLDM and EDM-SyCo, both trained on ZINC250K using the same synthetic coordinates and report the results in Table 2.

Table 2: Comparison between GeoLDM and EDM-SYCO on ZINC250K. To enable the training and evaluation of GeoLDM, we use the same synthetic coordinates as EDM-SYCO and use GeoLDM's bond type prediction algorithm.

|  | GEOLDM | EDM-SYCO |
|---|---|---|
| FCD ($\uparrow$) | 0.17 | **0.85** |
| KL ($\uparrow$) | 0.79 | **0.96** |
| NOVELTY ($\uparrow$) | **1.0** | **1.0** |
| UNIQUENESS ($\uparrow$) | **1.0** | **1.0** |
| VALIDITY ($\uparrow$) | 0.12 | **0.88** |

From the above discussion, we can conclude that (i) applying 3D models off-the-shelf as 2D baselines is either not directly possible (for datasets without 3D information) or yields limited performance due to the inability of the used bond type prediction algorithm to produce large and valid molecules, and (ii) combining EDM or GeoLDM with our proposed components would yield exactly our model since EDM and GeoLDM both share the same diffusion framework, which we also use in EDM-SyCo.

### 6.2 COMPARISON TO 2D MOLECULE GENERATIVE MODELS

In the following set of experiments, we compare the performance of EDM-SYCO to several autoregressive and to all-at-once diffusion-based baselines on de novo and conditional generation tasks as well as on a constrained optimization task. Specifically, comparing EDM-SYCO with the diffusion-based baselines highlights the benefit of defining the diffusion process on the latent 3D space compared to directly defining it on the 2D space.. More experiments are discussed in Appendix G. We use two datasets of different sizes and complexities: ZINC250K (Irwin et al., 2012) containing 250K molecules with up to 40 atoms, and GuacaMol (Brown et al., 2019) containing $\approx$1.5M drug-like molecules with up to 88 atoms. We train EDM-SYCO as described in Sections 3 and 4 on these two datasets. Training details and hyperparameters are given in Appendix F.

**De novo Generation** We leverage the GuacaMol benchmark (Brown et al., 2019), an evaluation framework for de novo molecular graph generation. We consider two metrics that measure the similarity between the generated molecules and the training set. The Fréchet ChemNet Distance (FCD) compares the hidden representations of ChemNet, a neural network capturing chemical and biological features of molecules (Preuer et al., 2018), and the KL divergence (KL) compares the probability distributions of a variety of physiochemical descriptors. We also report novelty, uniqueness, and validity scores, which measure the fraction of generated molecules that are not in the training set, that are pairwise different, and that are chemically valid according to RDKit, respectively.

Table 3: Evaluation metrics on the ZINC250K dataset. Results for methods with * are taken from their original papers. Methods that do no report the KL metric are denoted with N.A.. We split the methods into autoregressive and all-at-once methods. The **best scores within each category are in bold**. EDM-SyCO outperforms all all-at-once baselines by more than 26% on the FCD metric. Standard deviations are across 3 different training runs of the models.

| | METHOD | FCD (↑) | KL (↑) | NOVELTY (↑) | UNIQUENESS (↑) | VALIDITY (↑) |
|---|---|---|---|---|---|---|
| **AUTOREGRESSIVE** | JT-VAE (2018) | $0.75 \pm 0.01$ | $0.93 \pm 0.00$ | $\mathbf{1.00} \pm \mathbf{0.00}$ | $\mathbf{1.00} \pm \mathbf{0.00}$ | $\mathbf{1.00} \pm \mathbf{0.00}$ |
| | GRAPHAF (2020) | $0.05 \pm 0.00$ | $0.67 \pm 0.01$ | $0.91 \pm 0.01$ | $0.91 \pm 0.01$ | $\mathbf{1.00} \pm \mathbf{0.00}$ |
| | HIERVAE (2020) | $0.50 \pm 0.14$ | $0.92 \pm 0.00$ | $0.96 \pm 0.01$ | $0.96 \pm 0.01$ | $\mathbf{1.00} \pm \mathbf{0.00}$ |
| | MOLER (2021) | $\mathbf{0.83} \pm \mathbf{0.00}$ | $\mathbf{0.97} \pm \mathbf{0.00}$ | $0.99 \pm 0.00$ | $0.99 \pm 0.00$ | $\mathbf{1.00} \pm \mathbf{0.00}$ |
| | PS-VAE (2022) | $0.28 \pm 0.01$ | $0.84 \pm 0.00$ | $\mathbf{1.00} \pm \mathbf{0.00}$ | $\mathbf{1.00} \pm \mathbf{0.00}$ | $\mathbf{1.00} \pm \mathbf{0.00}$ |
| | GRAPHARM* (2023) | $0.04 \pm 0.00$ | N.A. | $\mathbf{1.00} \pm \mathbf{0.00}$ | $0.99 \pm 0.00$ | $0.88 \pm 0.00$ |
| | MICAM (2023) | $0.63 \pm 0.02$ | $0.94 \pm 0.00$ | $0.98 \pm 0.00$ | $0.98 \pm 0.00$ | $\mathbf{1.00} \pm \mathbf{0.00}$ |
| | MAGNET (2023) | $0.76 \pm 0.00$ | $0.95 \pm 0.00$ | $0.99 \pm 0.00$ | $0.99 \pm 0.00$ | $\mathbf{1.00} \pm \mathbf{0.00}$ |
| **ALL-AT-ONCE** | GDSS* (2022) | $0.10 \pm 0.01$ | N.A. | $\mathbf{1.00} \pm \mathbf{0.00}$ | $\mathbf{1.00} \pm \mathbf{0.00}$ | $0.97 \pm 0.01$ |
| | DIGRESS (2022) | $0.65 \pm 0.00$ | $0.91 \pm 0.00$ | $0.99 \pm 0.00$ | $0.99 \pm 0.00$ | $0.85 \pm 0.01$ |
| | GEOLDM (2023) | $0.17 \pm 0.00$ | $0.79 \pm 0.00$ | $\mathbf{1.00} \pm \mathbf{0.00}$ | $\mathbf{1.00} \pm \mathbf{0.00}$ | $0.12 \pm 0.00$ |
| | GRUM* (2023) | $0.64 \pm 0.01$ | N.A. | $\mathbf{1.00} \pm \mathbf{0.00}$ | $\mathbf{1.00} \pm \mathbf{0.00}$ | $\mathbf{0.99} \pm \mathbf{0.00}$ |
| | SWINGNN* (2023) | $0.67 \pm 0.00$ | N.A. | $0.96 \pm 0.00$ | $\mathbf{1.00} \pm \mathbf{0.00}$ | $0.91 \pm 0.00$ |
| | EDM-SYCO (OURS) | $\mathbf{0.85} \pm \mathbf{0.01}$ | $\mathbf{0.96} \pm \mathbf{0.00}$ | $\mathbf{1.00} \pm \mathbf{0.00}$ | $\mathbf{1.00} \pm \mathbf{0.00}$ | $0.88 \pm 0.01$ |

All scores, including FCD and KL, are normalized to lie between 0 and 1 and such that a higher score corresponds to a better performance. More details on these metrics are in Appendix F.8. De novo generation is a prerequisite for more complex tasks such as conditional generation and optimization.

Tables 3 and 4 show the performance of EDM-SyCO and the baselines on the ZINC250K and GuacaMol datasets, respectively. Notably, EDM-SyCO outperforms all all-at-once diffusion based models on the FCD metric by more than 26% on ZINC250K and 16% on GuacaMol. This shows the capacity of our model to more accurately capture the joint distribution of nodes and edges encoded in the latent features and coordinates. EDM-SyCO sets the new state-of-the-art FCD score on ZINC250K, outperforming all autoregressive and fragment-based baselines, which incorporate more domain knowl-

Table 4: FCD and KL scores on GuacaMol. * means corresponding numbers are from (Maziarz et al., 2021). DiGress results are from Vignac et al. (2022). For our model, we additionally report standard deviation across 3 validations of the same model.

| | | FCD (↑) | KL (↑) |
|---|---|---|---|
| **AUTOREG.** | CGVAE* (2018) | 0.26 | N.A. |
| | HIERVAE* (2020) | 0.62 | N.A. |
| | JT-VAE* (2018) | 0.73 | N.A. |
| | MOLER* (2021) | **0.81** | N.A. |
| **AAO** | DIGRESS (2022) | 0.68 | **0.93** |
| | EDM-SYCO (OURS) | $0.79 \pm 0.002$ | $0.93 \pm 0.006$ |

edge. Note that due to computational constraints, we use the same hyperparameters from ZINC250K to train our model on GuacaMol, while DiGress and MoLeR have been tuned for both datasets. Regarding validity, all autoregressive baselines achieve 100% validity scores due to valency checks after each intermediate step, which is usually not done for all-at-once methods. We show some sample molecules generated by our model in Figures 4 and 5 for the two datasets. We include an ablation study on the inductive bias of our architecture in Appendix G.10, compare to an EGNN-free encoder setting in Appendix G.11, and provide a runtime anaylsis in Appendix G.12.

**Property-Conditioned Generation**
To evaluate the conditional generation performance of our approach, we follow Ninniri et al. (2023) and condition the generation of 1000 molecules on LogP and molecular weight (MW) values sampled from ZINC250K. We

Table 5: Conditional generation results.

| METHOD | LOGP | | MW | |
|---|---|---|---|---|
| | MAE (↓) | VALID (↑) | MAE (↓) | VALID (↑) |
| DIGRESS | $0.49 \pm 0.05$ | 65% | $60.30 \pm 6.10$ | 73% |
| FREEGRESS | $0.15 \pm 0.02$ | **85%** | $15.18 \pm 2.71$ | 75% |
| EDM-SYCO | $\mathbf{0.14} \pm \mathbf{0.00}$ | 76% | $\mathbf{3.86} \pm \mathbf{0.08}$ | **88%** |

report the mean absolute error (MAE) between the target and estimated values for the generated molecules (using RDKit) and the validity rate. More details on the experimental setup are in Appendix F.6. We run the regressor guidance algorithm, discussed in Section 5. As baselines, we compare to DiGress, which uses a discrete guidance scheme, and FreeGress, which proposes a

Table 6: Similarity-constrained optimization results from (Jin et al., 2020). We split all learning-based baslines into translation and optimization approaches.

|  | METHOD | LogP (SIM $\geq 0.4$) | | LogP (SIM $\geq 0.6$) | | QED (SIM $\geq 0.4$) | | DRD2 (SIM $\geq 0.4$) | |
|---|---|---|---|---|---|---|---|---|---|
|  |  | IMPROV. | DIV. | IMPROV. | DIV. | SUCC. | DIV. | SUCC. | DIV. |
| TRANSL. | SEQ2SEQ | $3.37 \pm 1.75$ | 0.471 | $2.33 \pm 1.17$ | 0.331 | 58.5% | 0.331 | 75.9% | 0.176 |
| | JTNN | $3.55 \pm 1.67$ | 0.480 | $2.33 \pm 1.24$ | 0.333 | 59.9% | 0.373 | 77.8% | 0.156 |
| | ATOMG2G | $\mathbf{3.98} \pm \mathbf{1.54}$ | 0.563 | $2.41 \pm 1.19$ | 0.379 | 73.6% | 0.421 | 75.8% | 0.128 |
| | HIERG2G | $\mathbf{3.98} \pm \mathbf{1.46}$ | **0.564** | $\mathbf{2.49} \pm \mathbf{1.09}$ | **0.381** | **76.9%** | **0.477** | **85.9%** | **0.192** |
| OPTIM. | JT-VAE | $1.03 \pm 1.39$ | - | $0.28 \pm 0.79$ | - | 8.8% | - | 3.4% | - |
| | CG-VAE | $0.61 \pm 1.09$ | - | $0.25 \pm 0.74$ | - | 4.8% | - | 2.3% | - |
| | GCPN | $2.49 \pm 1.30$ | - | $0.79 \pm 0.63$ | - | 9.4% | 0.216 | 4.4% | 0.152 |
| | EDM-SYCO | $\mathbf{3.11} \pm \mathbf{1.27}$ | **0.559** | $\mathbf{1.51} \pm \mathbf{1.10}$ | **0.388** | **46.4%** | **0.352** | **27.3%** | **0.303** |

regressor-free guidane mechanism Ninniri et al. (2023). Table 5 shows the results. While EDM-SYCO achieves a comparable score to FreeGress on LogP, it outperforms it by a factor of 3.9 on MW. Compared to DiGress, it improves on LogP and MW by factors 3.5 and 15.6, respectively. This experiment highlights the benefit of our continuous latent space on conditional generation compared to operating directly on the discrete graph space. It also shows that graph-level properties are represented sufficiently well in this space. We include additional related experiments in Appendix G.4 and Appendix G.5.

**Similarity-Constrained Optimization** We follow Jin et al. (2020) and evaluate our optimization procedure from Section 5 with EDM-SYCO on their constrained optimization tasks: LogP, QED, and DRD2 on ZINC250K. These tasks consist of improving the properties of a set of 800 test molecules under the constraint that the Tanimoto similarity with Morgan fingerprints (Rogers & Hahn, 2010) between the optimized and initial molecule is above a given threshold. We report the average improvement for the LogP tasks and success rates for the binary tasks of QED and DRD2 consisting of translating molecules from a low range into a higher range. Additionally, we report the average diversity among the optimized molecules. More details are given in Appendix F.7.

We split the learning-based baselines from (Jin et al., 2020) into optimization and translation approaches. Translation approaches are specifically designed for such tasks and are trained on pairs of molecules exhibiting the improvement and similarity constraints, giving them an advantage over optimization methods. Since we leverage the same generative model and regressor from previous experiments without further training on this task, optimization approaches are the main point of comparison. Table 6 shows that EDM-SYCO outperforms all optimization baselines by an average factor of 3.5 across all tasks while achieving comparable performance to the translation approaches. These results further support the effectiveness of our SYCO framework and our novel constrained optimization algorithm. Figures 6 and 7 illustrate some of EDM-SYCO's optimized molecules. Thanks to our atom-based approach, the optimized molecules frequently only change in a few atoms from the starting molecule, which would be harder to achieve for fragment-based models.

## 7 CONCLUSION

Despite the common practice of using fragment-based autoregressive models for molecular graph generation, we demonstrated with SYCO that a mapping between molecular graphs and latent Euclidean point clouds enables atom-based all-at-once approaches to be competitive with special-purpose molecular graph generators. Further, we introduced a similarity-constrained optimization procedure for 3D diffusion models based on guidance and inpainting. Based on this framework, we developed EDM-SYCO, which sets a new state-of-the-art FCD score on ZINC250K. In our conditional generation and optimization experiments, we found our guidance-based approach to accurately guide the generation process, outperforming all baselines. With these results, we conclude that latent Euclidean generative models hold significant promise for advancing molecular graph generation and accelerating drug discovery. We discuss our work's broader impact in Appendix B.

## ACKNOWLEDGMENTS

This project is supported by the DAAD programme Konrad Zuse Schools of Excellence in Artificial Intelligence, sponsored by the Federal Ministry of Education and Research. Further, it is funded by the Federal Ministry of Education and Research (BMBF) and the Free State of Bavaria under the Excellence Strategy of the Federal Government and the Länder. This project is also supported by the Bavarian Ministry of Economic Affairs, Regional Development and Energy with funds from the Hightech Agenda Bayern.

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

# A    VISUALIZATIONS FOR GENERATED AND OPTIMIZED MOLECULES BY EDM-SYCO

We visualize some of the molecules generated by EDM-SYCO. For de novo generation, we show molecules of the model trained on ZINC250K in Figure 4, and of the one trained on GuacaMol in Figure 5. For the similarity-constrained optimization task, we show some molecules found by our optimization approach in the LogP task in Figure 6, and in the QED task in Figure 7.

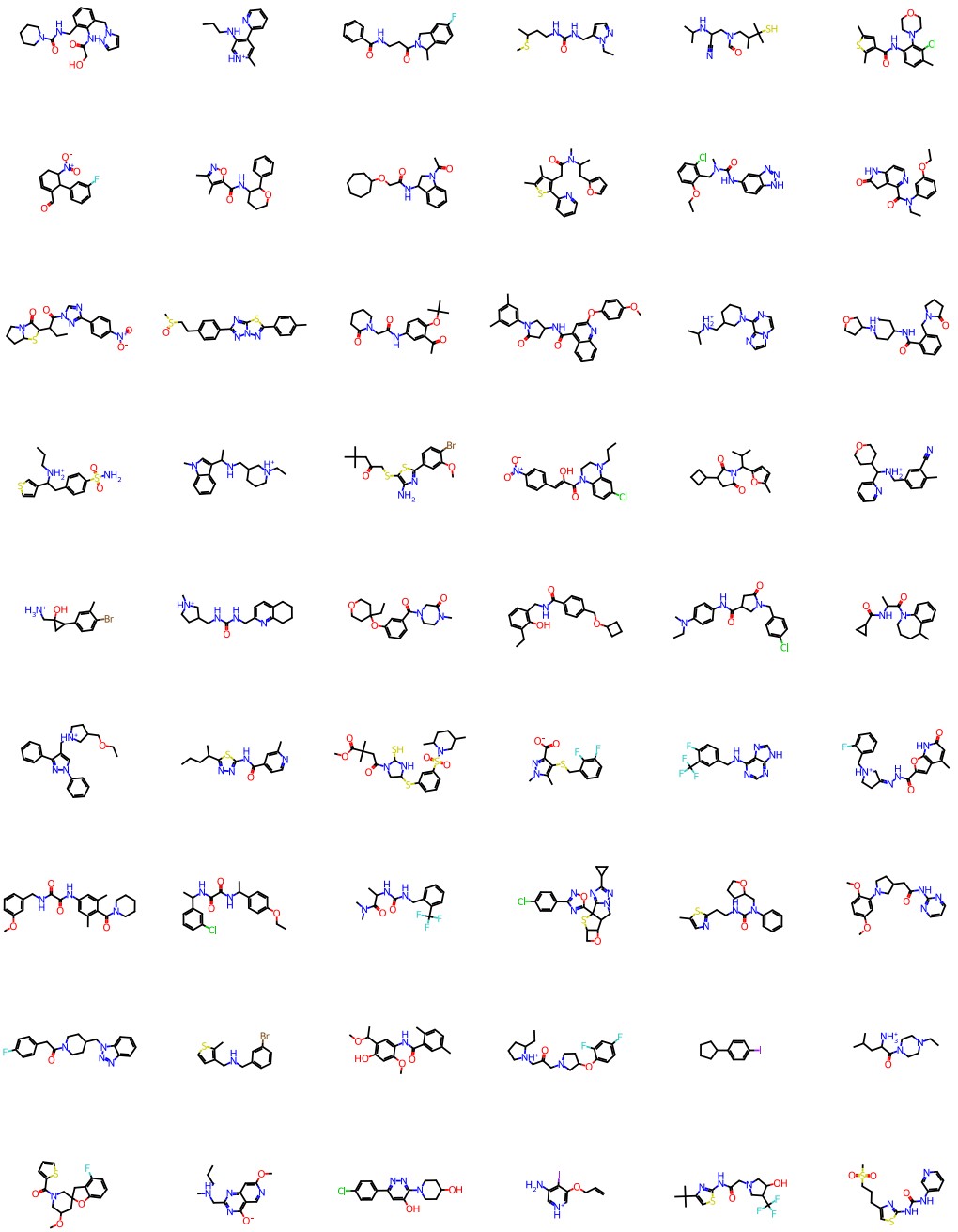

Figure 4: Sample molecules generated by EDM-SYCO trained on ZINC250K.

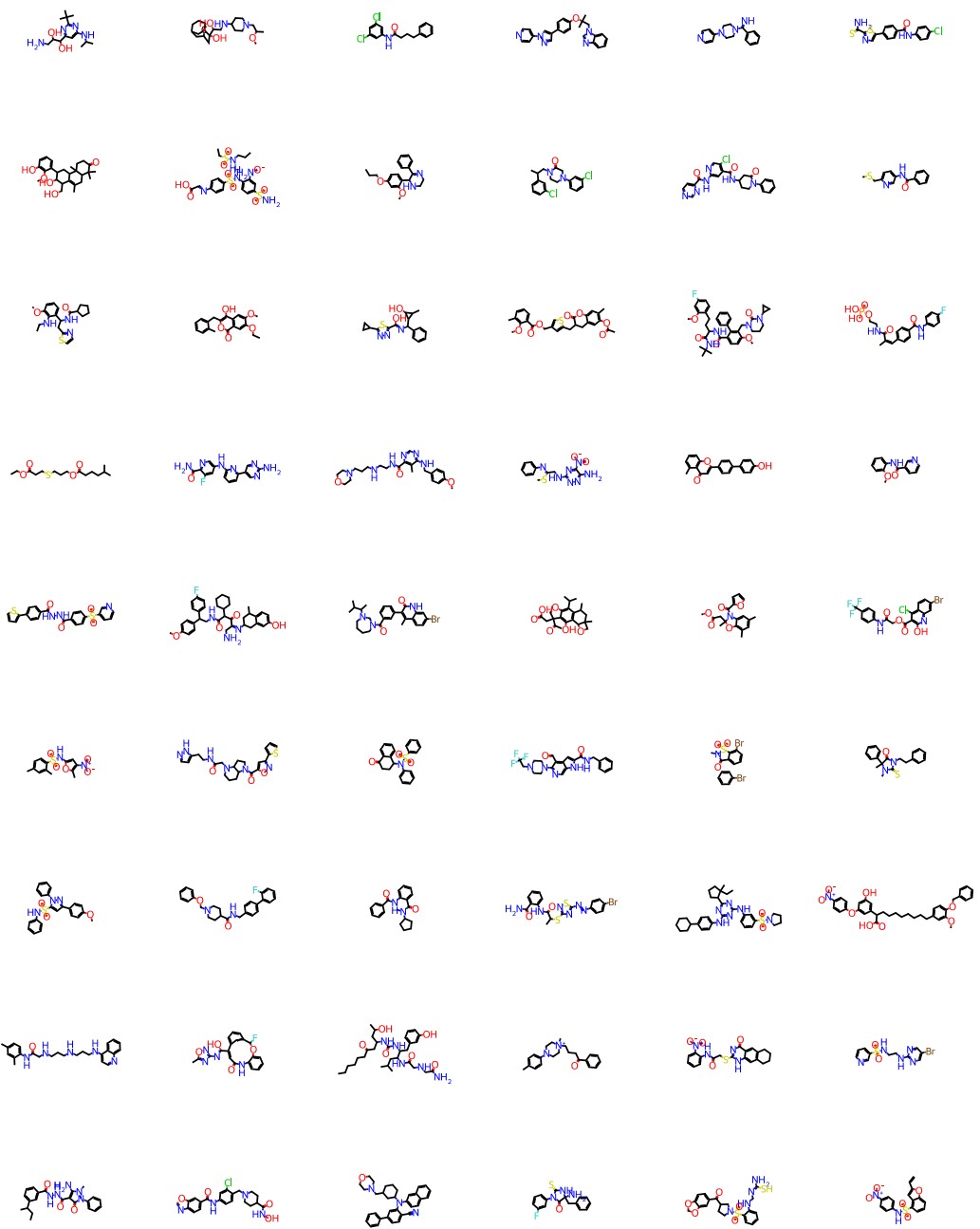

Figure 5: Sample molecules generated by EDM-SYCO trained on GuacaMol.

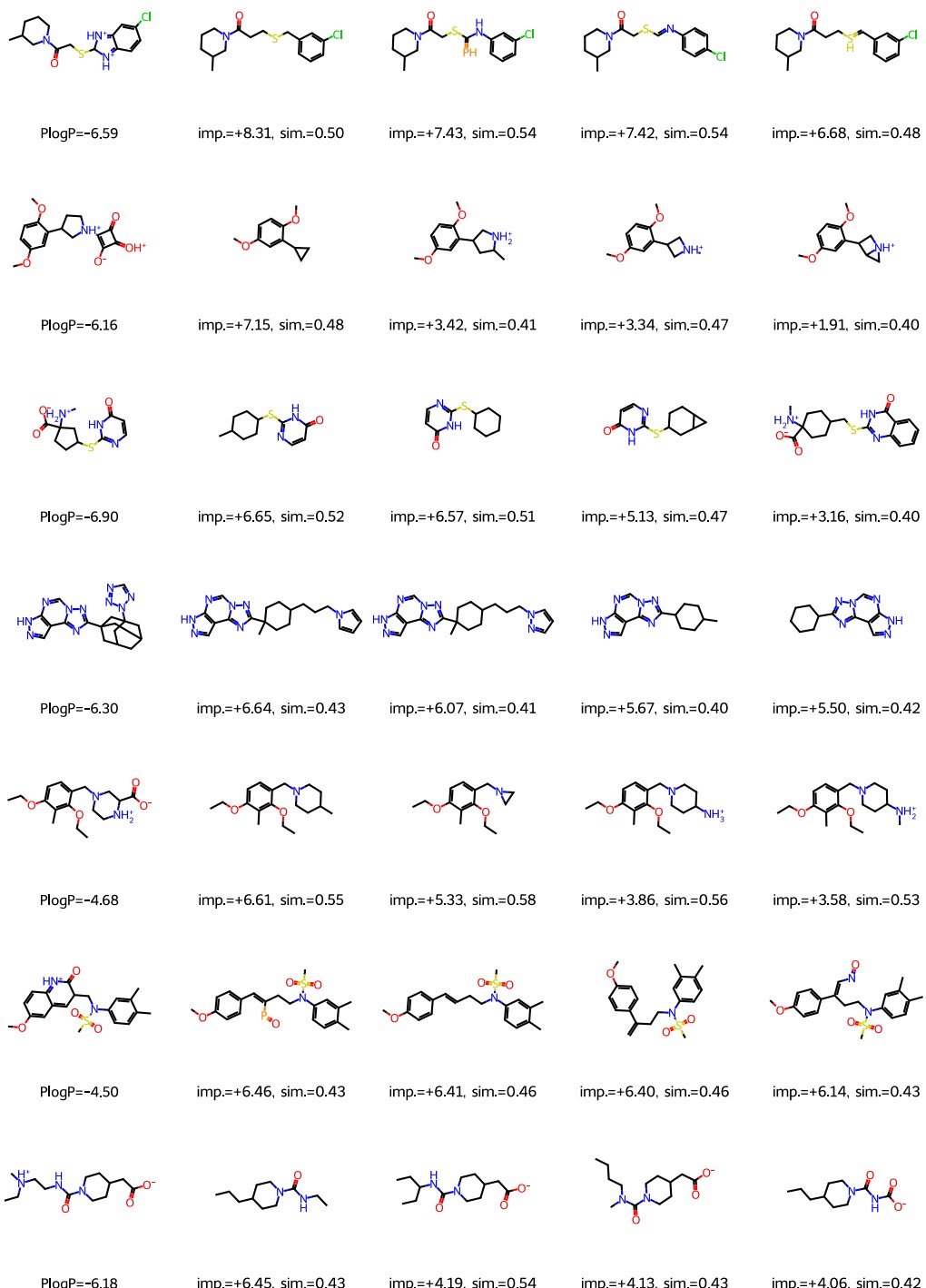

Figure 6: Molecules with the highest improvement in the LogP-constrained optimization task. Each row corresponds to one test molecule and 4 successful optimization results. We show the LogP value of the initial molecules, and for the optimized molecules, the achieved improvement (imp.) and the Tanimoto similarity to the initial molecule (sim.).

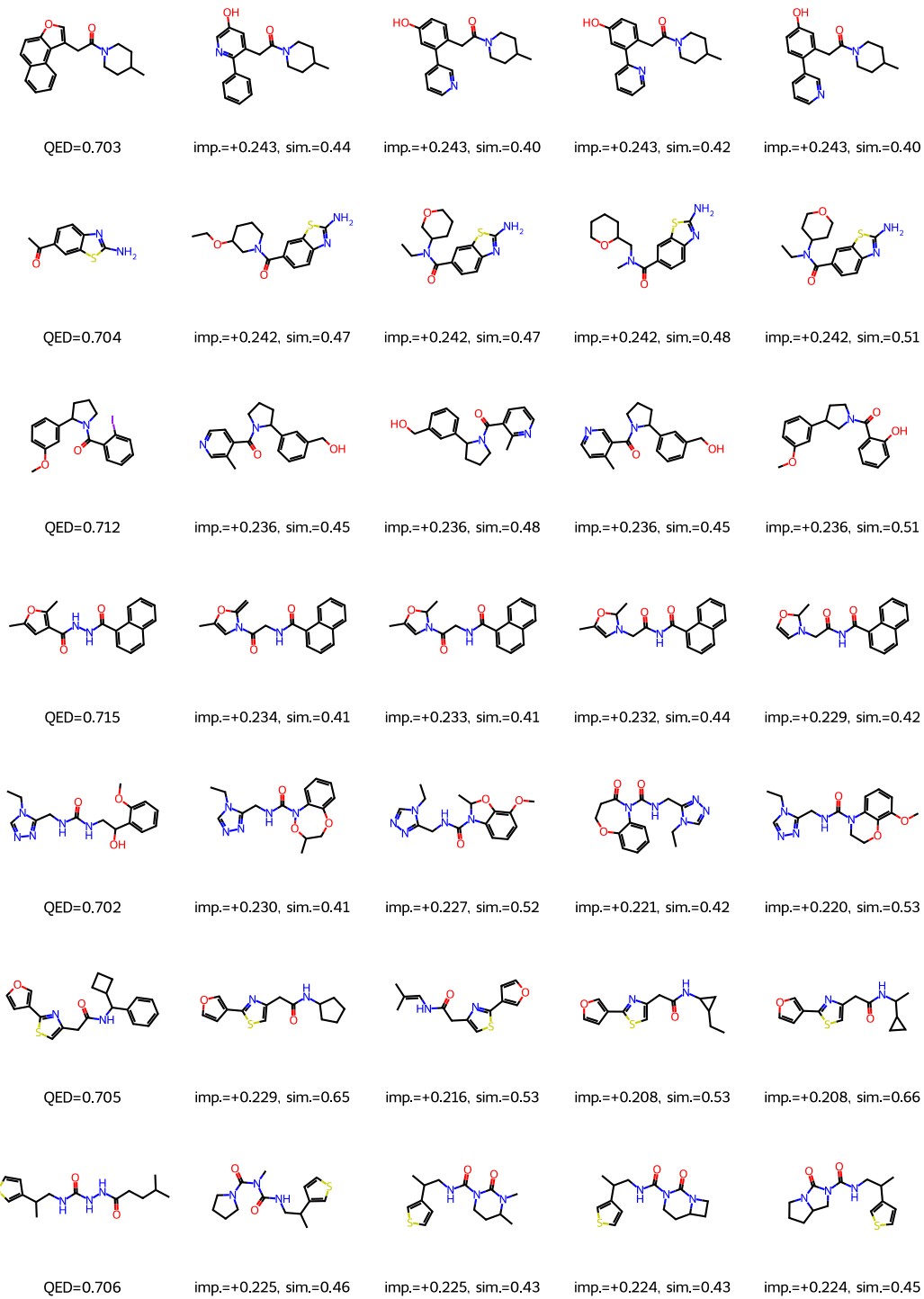

Figure 7: Molecules with highest improvement in the QED constrained optimization task. Each row corresponds to one test molecule and 4 successful optimization results. We show the QED value of the initial molecules, and for the optimized molecules, the achieved improvement (imp.) and the Tanimoto similarity to the initial molecule (sim.).

## B   Broader Impact

This paper advances the field of computational drug discovery by proposing new efficient methods for generative models. While there may be unintended consequences, such as the misuse of chemical weapons, we firmly believe that their benefits significantly outweigh the minuscule chance of misuse.

## C   Invariance and Equivariance

In this work, we are dealing with two types of symmetries arising from how we represent the data and/or physical symmetries. Specifically, graphs are invariant to permuting their nodes, meaning that a single graph of $N$ nodes can be represented by $N!$ different representations corresponding to $N!$ different orderings of its nodes. We refer to this transformation as the elements of the symmetric group $S_N$. Similarly, a point cloud representing a molecule in 3D can be arbitrarily rotated, reflected, and/or translated and still refer to the same molecule. This corresponds to an *infinite* number of different representations for the same object. We refer to this transformation as the action of the Euclidean group $E(3)$.

### C.1   $S_N$- and $E(3)$-Invariant Training Process

To efficiently learn from such data, we need to develop training methods invariant to these symmetries without needing data augmentation techniques. Concretely, we must ensure that the gradient updates used to train our model do not change if the molecular graph is permuted and/or the latent point cloud representation is rotated/reflected/translated. We require two ingredients to achieve this: an equivariant architecture and an invariant loss (Vignac et al., 2022). Based on the model architecture and the loss function described in the main text, we can show that EDM-SYCO satisfies both requirements and that its training process is, thus, invariant to the action of the permutation group $S_N$ and the Euclidean group $E(3)$.

As our ultimate goal is to generate molecular graphs, we provide more details on the permutation invariance properties of the distribution of molecular graphs learned by EDM-SYCO in the next section.

### C.2   Proof of Proposition 4.1 ($S_N$-Invariant Marginal Distribution)

In this section, we provide a formal proof for the statement from the main text that the molecular graph distribution defined by EDM-SYCO is invariant to permutations. For convenience, we provide the proposition again:

**Proposition C.1.** *The marginal distribution of molecular graphs $p_{\theta,\xi}(\mathcal{G}) = \mathbb{E}_{p_\theta(\boldsymbol{z}_0)}\left[p_\xi(\mathcal{G}|\boldsymbol{z}_0)\right]$ defined by the EDM and the decoder, is an $S_N$-invariant distribution, i.e. for any molecular graph $\mathcal{G} = (\boldsymbol{h}, \boldsymbol{A})$, $p_{\theta,\xi}(\boldsymbol{h}, \boldsymbol{A}) = p_{\theta,\xi}(P\boldsymbol{h}, P\boldsymbol{A}P)$ for any permutation matrix $P \in S_N$.*

**Note**: This invariance property is required for efficient likelihood computation, as the likelihood of a graph is the sum of the likelihoods of its $N!$ permutations, which is intractable to compute. However, when ensuring that all these permutations are assigned equal likelihood, it suffices to compute the likelihood of an arbitrary permutation.

*Proof.* The idea of the proof is when the initial distribution of the diffusion model $p_\theta(\boldsymbol{z}_T)$ is invariant and the transition distributions $p_\theta(\boldsymbol{z}_{t-1}|\boldsymbol{z}_t)$ are equivariant, then all marginal distributions at all diffusion time steps $p_\theta(\boldsymbol{z}_t)$ are invariant, including $p_\theta(\boldsymbol{z}_0)$ (Xu et al., 2022; 2023; Hoogeboom et al., 2022). With the same logic and with the equivariant decoder taking the place of the transition distribution, the graph distribution $p_{\theta,\xi}(\mathcal{G})$ will be invariant.

We prove this result by induction:

Base case: $p_\theta(\boldsymbol{z}_T) = \mathcal{N}(\boldsymbol{z}_T; \boldsymbol{0}, \boldsymbol{I})$ is permutation-invariant, i.e. $p_\theta(\boldsymbol{z}_T) = p_\theta(P\boldsymbol{z}_T)$ for all permutation matrices $P \in \{0,1\}^{N \times N}$ acting on $\boldsymbol{z}_T \in \mathbb{R}^{N \times (d+3)}$ by permuting its rows. This holds because all rows are i.i.d., as $p_\theta(\boldsymbol{z}_T)$ has a diagonal covariance matrix.

Induction step: Assume $p_\theta(z_t)$ is permutation-invariant. We show that if $p_\theta(z_{t-1}|z_t)$ is permutation-equivariant, i.e. $p_\theta(z_{t-1}|z_t) = p_\theta(Pz_{t-1}|Pz_t)$ (which holds when using a permutation-equivariant architecture $\epsilon_\theta$), then $p_\theta(z_{t-1})$ will be permutation-invariant.

$$
\begin{aligned}
p_\theta(Pz_{t-1}) &= \int_{z_t} p_\theta(Pz_{t-1}|z_t)p_\theta(z_t) \qquad \text{(Chain rule of probability)} \\
&= \int_{z_t} p_\theta(Pz_{t-1}|PP^{-1}z_t)p_\theta(PP^{-1}z_t) \qquad (PP^{-1} = I) \\
&= \int_{z_t} p_\theta(z_{t-1}|P^{-1}z_t)p_\theta(P^{-1}z_t) \qquad \text{(Equivariance and invariance)} \\
&= \int_u p_\theta(z_{t-1}|u)p_\theta(u) \underbrace{|\det P|}_{=1} \\
&\qquad \text{(Change of variables } u = P^{-1}z_t \text{ and } P \text{ is orhtogonal, so } \det P = \pm 1) \\
&= p_\theta(z_{t-1})
\end{aligned}
$$

By induction, $p_\theta(z_T), \ldots, p_\theta(z_0)$ are all permutation-invariant.

Finally, since the decoder $p_\xi(\mathcal{G}|z_0)$ is permutation-equivariant (as it applies the same operation on each node and each pair of nodes, thus not depending on the nodes ordering), with the same derivation, we also conclude that the final induced distribution $p_{\theta,\xi}(\mathcal{G})$ is permutation-invariant. $\qquad \square$

**Note**: A similar derivation can also be applied to show the invariance of $p_\theta(z_0)$ to rotations and reflections of the coordinates, as the only requirement on $P$ for the derivation to hold is to be orthogonal, which is the case for rotation and reflection matrices $R$.

## D  $E(n)$ EQUIVARIANT GRAPH NEURAL NETWORKS (EGNNS)

To perform equivariant updates in our latent space, we use the $E(n)$ Equivariant Graph Neural Network (EGNN) architecture (Satorras et al., 2021).

An EGNN operates on a point cloud of size $N$, where each point has features $h_i \in \mathbb{R}^d$ and coordinates $x_i \in \mathbb{R}^3$ associated with it. Let $h \in \mathbb{R}^{N \times d}$ and $x \in \mathbb{R}^{N \times 3}$ be the stacked features and coordinates, respectively, of all points. An EGNN is composed of Equivariant Graph Convolutional Layers (EGCLs) $h^{l+1}, x^{l+1} = \text{EGCL}(h^l, x^l)$. A single EGCL layer is defined as

$$m_{ij} = \phi_e(h_i^l, h_j^l, d_{ij}^2, a_{ij}) \tag{12}$$

$$h_i^{l+1} = \phi_h(h_i^l, \sum_{j \neq i} \tilde{e}_{ij} m_{ij}) \tag{13}$$

$$x_i^{l+1} = x_i^l + \sum_{j \neq i} \frac{x_i^l - x_j^l}{d_{ij} + 1} \phi_x(h_i^l, h_j^l, d_{ij}^2, a_{ij}), \tag{14}$$

where $l$ denotes the layer index, $d_{ij} = \|x_i^l - x_j^l\|_2$ is the Euclidean distance between points $i$ and $j$, and $a_{ij}$ are optional edge attributes which we set to $\|x_i^0 - x_j^0\|_2^2$. $\tilde{e}_{ij}$ serves as an attention mechanism that infers a soft estimation of the edges $\tilde{e}_{ij} = \phi_{inf}(m_{ij})$. Note that to update each point's features and coordinates, the EGCL layers consider all the other nodes, effectively treating the point cloud as a fully connected graph. All learnable components ($\phi_e, \phi_h, \phi_x$, and $\phi_{inf}$) are fully connected neural networks. An EGNN architecture is then composed of $L$ EGCL layers, denoted as $h^L, x^L = \text{EGNN}(h^0, x^0)$ with $h^0 = h$ and $x^0 = x$. The main hyperparameters of the EGNN architectures are the number of layers $L$ and a feature dimension $n_f$ used to control the width of the fully connected neural networks.

## E  ADDITIONAL METHOD DETAILS

In this section, we provide additional details for our method introduced in Sections 3, 4, and 5.

### E.1 TRAINING AND SAMPLING

We start by describing in more detail the training and sampling algorithms for EDM-SYCO in Algorithms 1 and 2.

---

**Algorithm 1** Training Algorithm

---

**Input:** a dataset of molecular graphs $\mathcal{G} = (\boldsymbol{h}, \boldsymbol{A})$
**Initial:** Encoder network $\mathcal{E}_\phi$, Decoder network $\mathcal{D}_\xi$, denoising network $\boldsymbol{\epsilon}_\theta$
**First Stage: Autoencoder Training**
**repeat**
    $\boldsymbol{\mu_z} \leftarrow \mathcal{E}_\phi(\boldsymbol{h}, \boldsymbol{A})$ {Encoder}
    $\boldsymbol{\epsilon_z} \sim \mathcal{N}(\boldsymbol{0}, \boldsymbol{I})$ and subtract center of mass from $\boldsymbol{\epsilon_z}^{(x)}$
    $\boldsymbol{z} \leftarrow \boldsymbol{\mu_z} + \sigma_0\boldsymbol{\epsilon_z}$
    $\hat{\boldsymbol{h}}, \hat{\boldsymbol{A}} \leftarrow \mathcal{D}_\xi(\boldsymbol{z})$
    $\mathcal{L}_{VAE}(\phi, \xi) \leftarrow \text{cross-entropy}\left(\begin{bmatrix} \hat{\mathbf{h}} \\ \hat{\mathbf{A}} \end{bmatrix}, \begin{bmatrix} \mathbf{h} \\ \mathbf{A} \end{bmatrix}\right)$
    Take an optimizer step on $\mathcal{L}_{AE}(\phi, \xi)$
**until** $\phi$ and $\xi$ have converged
**Second Stage: Diffusion Model Training**
Fix Autoencoder parameters $\phi$ and $\xi$
**repeat**
    $\boldsymbol{z}_0 \sim q_\phi(\boldsymbol{z}|\boldsymbol{h}, \boldsymbol{A})$
    $\boldsymbol{\epsilon_z} \sim \mathcal{N}(\boldsymbol{0}, \boldsymbol{I})$ and subtract center of mass from $\boldsymbol{\epsilon_z}^{(x)}$
    $t \sim \mathcal{U}(0, 1, \dots, T)$
    $\boldsymbol{z}_t \leftarrow \sqrt{\bar{\alpha}_t}\boldsymbol{z}_0 + \sqrt{1 - \bar{\alpha}_t}\boldsymbol{\epsilon_z}$
    $\mathcal{L}_{DM}(\theta) \leftarrow \|\boldsymbol{\epsilon_z} - \boldsymbol{\epsilon}_\theta(\boldsymbol{z}_t, t)\|^2$
    Take an optimizer step on $\mathcal{L}_{DM}(\theta)$
**until** $\theta$ has converged
**return** $\mathcal{E}_\phi, \mathcal{D}_\xi, \boldsymbol{\epsilon}_\theta$

---

**Algorithm 2** Sampling Algorithm

---

**Input:** Decoder network $\mathcal{D}_\xi$, denoising network $\boldsymbol{\epsilon}_\theta$
$\boldsymbol{z}_T \sim \mathcal{N}(\boldsymbol{0}, \boldsymbol{I})$ and subtract center of mass from $\boldsymbol{z}_T^{(x)}$
**for** $t = T$ **downto** 1 **do**
    $\boldsymbol{\epsilon_z} \sim \mathcal{N}(\boldsymbol{0}, \boldsymbol{I})$ and subtract center of mass from $\boldsymbol{\epsilon_z}^{(x)}$
    $\boldsymbol{z}_{t-1} \leftarrow \frac{1}{\sqrt{\alpha_t}}\left(\boldsymbol{z}_t - \frac{1-\alpha_t}{\sqrt{1-\bar{\alpha}_t}}\boldsymbol{\epsilon}_\theta(\boldsymbol{z}_t, t)\right) + \sigma_t\boldsymbol{\epsilon_z}$
**end for**
$\boldsymbol{h}, \boldsymbol{A} \leftarrow \mathcal{D}_\xi(\boldsymbol{z}_0)$
**return** 2D molecule $\mathcal{G} = (\boldsymbol{h}, \boldsymbol{A})$

---

### E.2 CONDITIONAL GENERATION

Here, we detail the regressor guidance algorithm we developed in Section 5. We showed in the main text that we can sample from the conditional distribution $q(\boldsymbol{z}_t|c)$ by replacing the predicted noise of the denoising neural network by Equation 7. By putting this back into the sampling equation, Equation 6, we get the following modified denoising step

$$\boldsymbol{z}_{t-1} = \boldsymbol{\mu}_\theta(\boldsymbol{z}_t, t) - \frac{1 - \alpha_t}{\sqrt{\alpha_t}}s\nabla_{\boldsymbol{z}_t}(g_\eta(\boldsymbol{z}_t) - c)^2 + \sigma_t\boldsymbol{\epsilon}_t. \tag{15}$$

A more general formulation for the new guidance term is given by $-s(t)\nabla_{\boldsymbol{z}_t}l(\boldsymbol{z}_t, c)$, where $s(t)$ is a time-dependent scaling function and $l$ can be any invariant loss function. In the specific case that we derived, $s(t) = \frac{1-\alpha_t}{\sqrt{\alpha_t}}s$ and $l(\boldsymbol{z}_t, c) = (g_\eta(\boldsymbol{z}_t) - c)^2$. $l$ can be replaced by any other loss function that we want to optimize, or in the case of property maximization, it can be the prediction of the regressor

directly (Equation 8). In our experiments, we found a simple linear schedule $s(t) = at$ to perform well, where $a \in \mathbb{R}$ is a hyperparameter that we tune for each task.

To ensure the equivariance of the distribution even with guidance, the new term $s(t)\nabla_{\boldsymbol{z}_t} l(\boldsymbol{z}_t, c)$ needs to be $E(3)$-equivariant. This is done by subtracting its center of mass at each sampling step and by choosing an invariant loss function $l$, as the differential operator of an invariant function yields an equivariant function (Chmiela et al., 2017; Schütt et al., 2017).

### E.3 Scaffold-constrained Generation

Here, we provide additional details for the inpainting algorithm described in Section 5, where we described an approach that allows to generate a point cloud that contains a chosen scaffold.

Lugmayr et al. (2022) observed that the version of their algorithm described in the main text leads to inconsistent completions and proposes to run each sampling step multiple times to allow the model to harmonize its generations. Concretely, each denoising step is repeated $r$ times to improve the generations' quality and allow the model to harmonize the unknown part to the scaffold. This is achieved by applying the forward diffusion model (Equation 2) after each denoising step. Lugmayr et al. (2022) also propose to go more than one forward step with the diffusion model to allow the model to better harmonize its generations. This choice is fixed by the jump length $j$. Note that this procedure increases the computation cost of running the model $r$ times, which might be a limiting factor. For this reason, we combine this approach with the approach described in Appendix G.9 that allows to reduce the number of reverse diffusion steps to run.

The procedure described in the main text gives a way to condition the diffusion model, which operates on the latent point cloud space. However, we are ultimately interested in generating a molecular graph containing a scaffold defined by a subset of atoms and the bonds between them. We map the molecular scaffold to its point cloud representation using our encoder to get $m \odot \boldsymbol{z}_0$. This enables running the inpainting algorithm described above. Then, we decode the sampled 3D point cloud to a molecular graph. Since our decoder model operates on each node and edge separately, the final molecular graph is guaranteed to contain the original scaffold as long as the scaffold in isolation can be correctly reconstructed with our autoencoder.

## F Implementation Details

### F.1 Licenses for used code, models, and datasets

We start by listing the assets we used in our work and their respective licenses. We use the following tools, codebases, and datasets:

- RDKit (Landrum et al., 2006) (BSD 3-Clause License)
- PyTorch (Paszke et al., 2019) (BSD 3-Clause License)
- EDM (Hoogeboom et al., 2022) (MIT license)
- GeoLDM (Xu et al., 2023) (MIT license)
- ZINC250K dataset (Irwin et al., 2012) (Custom license)
- GuacaMol dataset (Brown et al., 2019) (MIT license)

### F.2 Architecture Details

We train two sets of models with the same architectures on the two used datasets, ZINC250K and GuacaMol. The EDM, together with the autoencoder, has 9.2M total parameters, while the regressor has 4.2M parameters.

**Encoder.** The first part of the encoder $\mathcal{E}$, introduced in Section 3, is the conformer generation method, while the second part is an EGNN with 1 layer and 128 hidden features (See Appendix D). We found that having at least 1 EGNN layer is crucial to achieve high reconstruction accuracy for the autoencoder, and as this allows us to reach more than 99% accuracy, we did not use more layers. This EGNN takes as input a point cloud with the one-hot encodings of the atom types and formal charges

as features $h$ and the synthetic coordinates computed by the conformer generation part as coordinates $x$. The output is a processed point cloud with continuous node embeddings of dimension $d = 2$ and updated coordinates.

**Decoder.** The decoder $\mathcal{D}$, also introduced in Section 3, consists of 2 fully connected networks $\text{MLP}_{node}$ and $\text{MLP}_{edge}$, both having 256 hidden dimensions.

**EDM.** The denoising network of EDM is parametrized with an EGNN with 9 layers and 256 hidden features. The EDM has $T = 1000$ diffusion steps.

**Regressor.** The regressor used for the conditional generation and optimization experiments (see Section 6.2) is also an EGNN with 4 layers and 256 hidden features, followed by a sum pooling operation over the node features and then a fully connected network with 2 layers and a hidden dimension of 256 that maps the pooled graph embedding to the target values of the properties.

## F.3 TRAINING DETAILS

We use the original train/validation/test splits of the used datasets (Irwin et al., 2012; Brown et al., 2019). The autoencoder is trained in the first stage to minimize the cross-entropy loss between the ground truth and predicted graphs for a maximum of 100 epochs, with early stopping if the validation accuracy does not improve for 10 epochs. To deal with the class imbalance problem caused by the sparsity of the graph adjacency matrices and the dominance of some atom types and formal charge values, we scale each term of the cross-entropy loss with a class-specific value based on the statistics of the training set (King & Zeng, 2001). In the second training stage, the EDM model is trained for 1000 epochs on ZINC250 and approximately 300 epochs on GuacaMol.

The regressor is trained with the $L_1$ loss to predict the molecular properties on the noisy latent point cloud representations of the molecules by applying the same noise schedule of the EDM. The regressor is trained for 500 epochs, with early stopping if the MAE on the validation set does not improve after 50 epochs.

All models are trained with a batch size of 64 and using the recent Prodigy optimizer (Mishchenko & Defazio, 2023) with $d_{coef} = 0.1$, which we found to be a very important hyperparameter for the stability of training.

We train all models on ZINC250K on a single Nvidia A100 GPU, and on GuacaMol, we use multi-GPU training on 4 Nvidia A100 GPUs.

## F.4 DATASET DETAILS

We use two datasets in our experiments: ZINC250K (Irwin et al., 2012) and GuacaMol (Brown et al., 2019). For both datasets, we represent a molecule as a graph. The nodes are atoms with the one-hot encoding of their atom type and their formal charge as features. In addition to modeling all heavy atoms, we also model explicit H atoms, which we found to increase the validity of the generated molecules as it reduces some kekulization errors produced by rdkit [2]. The edges are chemical bonds with the one-hot encoding of the bond type as features. We model the following bond types: no-bond, single, double, triple, and aromatic. We model the absence of a bond as a separate bond type to allow the autoencoder to reconstruct the full bond information. These graphs present very unbalanced scales for the atom types, formal charges, and bond types. We use class-specific weights to train the autoencoder, as outlined in Section 3.

ZINC250K has a total of 250K molecules and 220,011 training molecules. It has 10 atom types (including H) and 3 formal charge values (-1, 0, 1), and the largest molecule has 40 atoms.

GuacaMol has a total of ≈1.5M molecules, of which we use 1,273,104 for training. It has 13 atom types (including H) and 5 formal charge values (-1, 0, 1, 2, 3), and the largest molecule has 88 atoms.

We compute the synthetic coordinates needed for our model in a preprocessing step for both datasets and use the computed coordinates throughout the training.

---

[2]https://github.com/rdkit/rdkit/wiki/FrequentlyAskedQuestionscant-kekulize-mol

### F.5 SAMPLING THE NUMBER OF ATOMS

In this work, the diffusion generative model operates on a fixed-size point cloud $N$. During sampling, the first step is to sample this number $N$. To sample molecules of similar sizes to those in the training set, we approximate the distribution of the number of nodes using a categorical distribution.

To this end, we count the number of occurrences of each molecule size in the training set and use the categorical distribution parameterized by the normalized counts to sample the number of atoms of generated molecules. This simple procedure works well for the task of unconditional generation. However, in the property-conditioned generation, we must consider that some properties are highly correlated to molecule size, such as molecular weight (MW). Therefore, for this task, we start by dividing the range of property values from the training set into 100 equal-sized bins. We define a different categorical distribution for each bin that counts the sizes of molecules falling in that bin. Given the conditioning value, we first specify the bin it falls into and use the corresponding categorical distribution to sample the number of nodes. Finally, for molecule optimization, we curate a dataset of molecule pairs that exhibit the desired property improvement and similarity constraints and create different conditional categorical distributions, one for each size of the initial molecule.

Note that these different procedures are needed due to the inherent limitation of diffusion models on graphs that require the number of nodes to be known and fixed in advance. Fundamentally, solving this issue would be a valuable contribution to the field, and we leave it for future work.

### F.6 PROPERTY-CONDITIONAL GENERATION EXPERIMENT

Following Ninniri et al. (2023), for each task, we sample 100 molecules from the test set and condition on their property values to generate 10 valid molecules each and report the mean absolute error between the target and the estimated values for the generated molecules measured by RDKit. We experimented with different values of the guidance scale $s$ and with using the $l_1$ and $l_2$ losses to compute the guidance loss but found the $l_2$ loss with scale values $s = 2.5$ and $s = 1.0$ for LogP and MW, respectively, to work best.

### F.7 CONSTRAINED MOLECULE OPTIMIZATION

Here, we provide more details on the constrained optimization tasks described in Section 6. Given a molecule $\mathcal{G}$, these tasks aim to find a different molecule $\mathcal{G}'$ with higher property values satisfying the similarity constraint $sim(\mathcal{G}, \mathcal{G}') \geq 0.4$ (or 0.6), where $sim$ computes the Tanimoto similarity with Morgan fingerprints (Rogers & Hahn, 2010). For the LogP task, the new molecule $\mathcal{G}'$ needs to have a higher LogP value and we report the average improvement over 800 molecules from the test set with the lowest LogP values. For the QED task, the goal is to optimize another 800 molecules from the test with QED values in the low range $[0.7, 0.8]$ into the high range $[0.9, 1.0]$, and we report the success rate over these molecules. For DRD2, the goal is to translate inactive compounds ($p \leq 0.05$) into active ones ($p \geq 0.5$), and we similarly report the success rate over these molecules. In addition, for all tasks, we report the diversity as the average pairwise distance between the successfully optimized molecules for each initial test molecule. The distance is defined as $dist(\mathcal{G}, \mathcal{G}') = 1 - sim(\mathcal{G}, \mathcal{G}')$.

**Choosing $t$.** As described in Section 5, the intermediate time step $t$ controls the trade-off between the similarity to the initial molecule and the capacity to improve its target properties. In principle, for each test molecule, there is an optimal $t$ yielding the highest improvement under the similarity constraints. However, finding the optimal $t$ for each single molecule is computationally expensive. Therefore, we use a subset of 20 molecules and run our optimization algorithm for different values of $t$ on these molecules to find the value of $t$ that yields the highest average improvement under the considered similarity constraints. We use this value for all test molecules. Our experiments found that a value of $t$ in the range $[500, 700]$ usually works best with the total $T = 1000$. Figure 8 shows the effect of the time step $t$ on the similarity and property value for one molecule from the test set. Table 7 lists the exact value of $t$ used for each task.

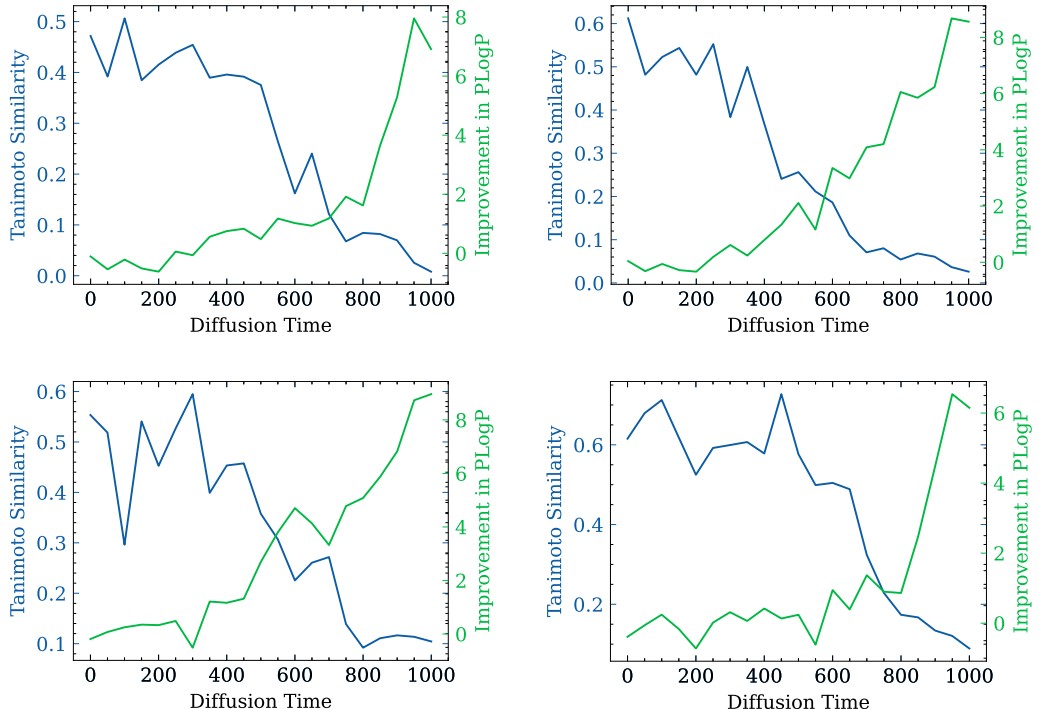

Figure 8: Effect of using different time steps $t$ on the similarity and improvement values in the similarity-constrained optimization task of PLogP. We visualize 4 different molecules from the test set.

Table 7: Values of the time step $t$ used in all optimization tasks.

|  | TIME STEP $t$ |
| --- | --- |
| PLOGP (SIM $\geq 0.4$) | 600 |
| PLOGP (SIM $\geq 0.6$) | 500 |
| QED | 650 |
| DRD2 | 650 |

## F.8 EVALUATION DETAILS

We provide additional details about the computation of the FCD and KL metrics (Brown et al., 2019) and additional insights into the novelty and uniqueness metrics. Additionally, we provide a detailed anaylsis of the validity rate of our model.

- **KL**: The following descriptors are computed for the generated and reference molecules: BertzCT, MolLogP, MolWt, TPSA, NumHAcceptors, NumHDonors, NumRotatableBonds, NumAliphaticRings, NumAromaticRings, and similarity to nearest neighbor with ECFP4 fingerprints. The KL divergence $D_{KL,i}$ is computed for each descriptor between the two sets of molecules and aggregated to a final normalized score via $\frac{1}{k}\sum_{i=1}^{k}\exp(-D_{KL,i})$.
- **FCD**: The FCD is computed based on the hidden representations of molecules in ChemNet, trained for predicting biological activities, similarly to the FID usually applied to image generative models. Concretely, the means and covariances of the last hidden activations of ChemNet are computed for the reference and generated molecules and the Frechet distance between them is computed. This distance is then normalized via $\exp(-0.2 \cdot FCD)$ to lie between 0 and 1.

- **Novelty & Uniqueness**: Interestingly, most of the methods used in this work achieve a novelty and uniqueness scores close to 100% on the two datasets ZINC250K and GuacaMol. These high scores are mainly due to the larger molecular sizes in our datasets compared to smaller datasets such as QM9. The number of possible molecules grows exponentially with the molecule size, making it increasingly unlikely to sample exact replica of the training molecules or sample the exact molecule twice (note than even if a single atom or bond type is different, the molecules are considered different).

- **Validity**: We analyze the invalid molecules generated by our model and find the following common errors raised by RDKit: Kekulization errors ( 55%) [3], Valency errors ( 33%) [4], A non-ring atom is marked aromatic ( 12%).

  All these errors are due to inconsistent bond type predictions for a given molecule. Because the autoencoder achieves a near-perfect reconstruction accuracy on the training and test sets, this inconsistency is likely due to a mismatch between the training distribution and the generated 3D molecules distribution. This could be improved by using more powerful 3D generative models, or one could try to jointly train the autoencoder and the latent diffusion model to adapt the decoder to the distribution of generated molecules.

## G    ADDITIONAL EXPERIMENTS

In this section, we provide additional experiments to showcase the performance of our model and analyze some of its properties.

### G.1    BOND TYPE PREDICTION FROM SYNTHETIC COORDINATES

A crucial part of successfully applying our method is to be able to accurately reconstruct the molecular graph from the Euclidean point cloud. This section provides an ablation study to compare our approach to other methods.

Formally, we are given a point cloud with features $h \in \{0, 1\}^{N \times (a+c)}$ containing the atom type and formal charge of each atom and coordinates $x \in \mathbb{R}^{N \times 3}$ computed using the conformer generation algorithm ETKDG (Riniker & Landrum, 2015). The goal is to reconstruct the bond types stored in the adjacency matrix $A \in \{0, 1\}^{N \times N \times b}$. We test the following approaches:

- **Rule-based**: This approach is widely used by generative models for molecules in 3D (Hoogeboom et al., 2022; Xu et al., 2023) and consists of using typical bond type lengths to infer the bond types between two atoms based on their features and the inter-atomic distances between them.[5] It uses a set of if-else statements and predicts a specific bond type if the length falls within a specific range that takes the features of the two atoms into account.

- **MLP**: We develop this baseline to generalize the lookup table approach and replace the set of if-else statements by an MLP. Formally, we construct edge features between atoms $i$ and $j$ as $[h_i, h_j, d_{ij}]$, where $d_{ij}$ is the Euclidean distance between the two atoms. This is also the same information that the lookup table method above uses to infer the bond types. Instead, we train an MLP to predict the bond type from this edge feature in a classification setting. Note that this MLP is defined on the edge level, so a molecule with $N$ atoms provides $N^2$ training examples for the MLP to learn from. Also note that we do not need to have an extra MLP for the atom type as they are directly available in this case since we do not run an EGNN in the first step.

- **EGNN + MLP**: The major limitation of the above approaches is that to predict the bond type between two atoms they only consider the distance between them and their respective features, without considering other atoms. However, not all molecules exhibit typical bond lengths between their atoms (Hoogeboom et al., 2022). To overcome this limitation, we combine the MLP approach with an EGNN that first computes updated coordinates and

---

[3]https://github.com/rdkit/rdkit/wiki/FrequentlyAskedQuestionscant-kekulize-mol

[4]https://github.com/rdkit/rdkit/wiki/FrequentlyAskedQuestionsexplicit-valence-for-atom–is-greater-than-permitted

[5]https://www.wiredchemist.com/chemistry/data/bond_energies_lengths.html

atom features considering the whole graph before running the local MLP to predict the bond type. Here, we add an extra MLP to predict the node labels (type and formal charge). This is the same architecture presented in the main text.

We evaluate these three approaches on ZINC250K using the synthetic coordinates computed as outlined in Section 3. We report the molecule reconstruction accuracy on the validation set, where a molecule is considered correctly reconstructed if all of its atom features (atom types and formal charges) and bond types are correctly reconstructed. Table 1 shows that the EGNN approach achieves an almost perfect reconstruction accuracy. This shows that (i) the synthetic coordinates contain all the needed information to reconstruct the bond types, which might not be directly clear if we are limited to the other methods, and (ii) incorporating the global graph information into the prediction task is crucial to accurately reconstruct the bonds. We adopt the EGNN approach into our model.

## G.2 CONFORMER GENERATION COMPARISON

To analyze the effect of different conformer generation algorithms on the performance of our model, we perform an ablation study using 2 different algorithms/packages from RDKit to compute the synthetic coordinates: the ETKDG algorithm and the Pharm3D package. We perform two training runs using the same setup except for the conformer generation method. For these two runs, we use smaller models (we halve the number of layers) than those reported in the main text and train for 500 epochs, so they are not directly comparable. Results are shown in Table 8.

Table 8: Model performance on ZINC using different conformer generation methods from RDKit.

|  | ETKDG | PHARM3D |
|---|---|---|
| VAE RECONSTRUCTION ACCURACY | 99.92% | 99.90% |
| KL | 0.94 | 0.93 |
| FCD | 0.75 | 0.70 |

## G.3 COMPLETE BENCHMARK RESULTS

We provide complete benchmark results for the GuacaMol benchmark (Brown et al., 2019) and the MOSES benchmark (Polykovskiy et al., 2020) in Tables 9 and 10, respectively. In addition to the baselines reported in the main text, we add two SMILES-based baselines and we add different flavours of MoLeR with different vocabulary sizes, denoted as MoLeR-$V$, where $V$ denotes the vocabulary size.

Table 9: GuacaMol benchmark metrics on the ZINC250K dataset. We report mean and standard deviations across 3 random training runs for all methods and metrics.

| METRIC | FCD ($\uparrow$) | KL ($\uparrow$) | NOVELTY ($\uparrow$) | UNIQUENESS ($\uparrow$) | VALIDITY ($\uparrow$) |
|---|---|---|---|---|---|
| CHARVAE | $0.17 \pm 0.08$ | $0.78 \pm 0.04$ | $\mathbf{0.99 \pm 0.00}$ | $0.99 \pm 0.00$ | $0.09 \pm 0.01$ |
| SMILES-LSTM | $\mathbf{0.93 \pm 0.00}$ | $\mathbf{1.00 \pm 0.00}$ | $0.98 \pm 0.00$ | $\mathbf{1.00 \pm 0.00}$ | $\mathbf{0.96 \pm 0.01}$ |
| GRAPHAF | $0.05 \pm 0.00$ | $0.67 \pm 0.01$ | $0.91 \pm 0.01$ | $0.91 \pm 0.01$ | $\mathbf{1.00 \pm 0.00}$ |
| HIERVAE | $0.50 \pm 0.14$ | $0.92 \pm 0.00$ | $0.96 \pm 0.01$ | $0.96 \pm 0.01$ | $\mathbf{1.00 \pm 0.00}$ |
| MICAM | $0.63 \pm 0.02$ | $0.94 \pm 0.00$ | $0.98 \pm 0.00$ | $0.98 \pm 0.00$ | $\mathbf{1.00 \pm 0.00}$ |
| JTVAE | $0.75 \pm 0.01$ | $0.93 \pm 0.00$ | $\mathbf{1.00 \pm 0.00}$ | $\mathbf{1.00 \pm 0.00}$ | $\mathbf{1.00 \pm 0.00}$ |
| PSVAE | $0.28 \pm 0.01$ | $0.84 \pm 0.00$ | $\mathbf{1.00 \pm 0.00}$ | $\mathbf{1.00 \pm 0.00}$ | $\mathbf{1.00 \pm 0.00}$ |
| MAGNET | $0.76 \pm 0.00$ | $0.95 \pm 0.00$ | $0.99 \pm 0.00$ | $0.99 \pm 0.00$ | $\mathbf{1.00 \pm 0.00}$ |
| MOLER-5 | $0.40 \pm 0.01$ | $0.93 \pm 0.01$ | $0.99 \pm 0.00$ | $0.97 \pm 0.00$ | $\mathbf{1.00 \pm 0.00}$ |
| MOLER-350 | $0.80 \pm 0.01$ | $\mathbf{0.98 \pm 0.00}$ | $\mathbf{1.00 \pm 0.00}$ | $\mathbf{1.00 \pm 0.00}$ | $\mathbf{1.00 \pm 0.00}$ |
| MOLER-2000 | $\mathbf{0.83 \pm 0.00}$ | $0.97 \pm 0.00$ | $0.99 \pm 0.00$ | $0.99 \pm 0.00$ | $\mathbf{1.00 \pm 0.00}$ |
| DIGRESS | $0.65 \pm 0.00$ | $0.91 \pm 0.00$ | $0.99 \pm 0.00$ | $0.99 \pm 0.00$ | $0.85 \pm 0.01$ |
| EDM-SYCO | $\mathbf{0.85 \pm 0.01}$ | $\mathbf{0.96 \pm 0.00}$ | $\mathbf{1.00 \pm 0.00}$ | $\mathbf{1.00 \pm 0.00}$ | $0.88 \pm 0.01$ |

Table 10: MOSES benchmark metrics on the ZINC250K dataset. We report mean and standard deviations across 3 random training runs for all methods and metrics.

| METRIC | FILTERS ($\downarrow$) | INTDIV ($\uparrow$) | INTDIV2 ($\uparrow$) | QED ($\downarrow$) | SA ($\downarrow$) | LOGP ($\downarrow$) | WEIGHT ($\downarrow$) |
|---|---|---|---|---|---|---|---|
| CHARVAE | $0.43 \pm 0.02$ | $0.88 \pm 0.01$ | $0.87 \pm 0.01$ | $0.06 \pm 0.03$ | $0.48 \pm 0.13$ | $0.87 \pm 0.14$ | $34.23 \pm 07.05$ |
| SMILES-LSTM | $0.59 \pm 0.01$ | $0.87 \pm 0.00$ | $0.86 \pm 0.00$ | $0.00 \pm 0.00$ | $0.04 \pm 0.02$ | $0.12 \pm 0.01$ | $02.62 \pm 00.16$ |
| GRAPHAF | $0.47 \pm 0.03$ | $0.93 \pm 0.00$ | $0.91 \pm 0.00$ | $0.22 \pm 0.01$ | $0.88 \pm 0.10$ | $0.41 \pm 0.02$ | $96.93 \pm 04.99$ |
| HIERVAE | $0.58 \pm 0.05$ | $0.87 \pm 0.01$ | $0.86 \pm 0.01$ | $0.03 \pm 0.00$ | $0.19 \pm 0.17$ | $0.35 \pm 0.21$ | $18.97 \pm 10.43$ |
| MICAM | $0.54 \pm 0.02$ | $0.87 \pm 0.00$ | $0.87 \pm 0.00$ | $0.08 \pm 0.00$ | $0.51 \pm 0.03$ | $0.20 \pm 0.05$ | $51.19 \pm 06.58$ |
| JTVAE | $0.57 \pm 0.00$ | $0.86 \pm 0.00$ | $0.86 \pm 0.00$ | $0.01 \pm 0.00$ | $0.35 \pm 0.01$ | $0.29 \pm 0.01$ | $02.93 \pm 00.07$ |
| PSVAE | $0.85 \pm 0.01$ | $0.89 \pm 0.00$ | $0.88 \pm 0.00$ | $0.05 \pm 0.00$ | $1.16 \pm 0.06$ | $0.34 \pm 0.01$ | $38.26 \pm 02.74$ |
| MAGNET | $0.70 \pm 0.00$ | $0.87 \pm 0.00$ | $0.87 \pm 0.00$ | $0.01 \pm 0.00$ | $0.13 \pm 0.00$ | $0.23 \pm 0.01$ | $14.52 \pm 00.40$ |
| MOLER-5 | $0.57 \pm 0.01$ | $0.86 \pm 0.00$ | $0.85 \pm 0.00$ | $0.01 \pm 0.00$ | $0.11 \pm 0.04$ | $0.17 \pm 0.10$ | $10.67 \pm 04.13$ |
| MOLER-350 | $0.56 \pm 0.00$ | $0.87 \pm 0.00$ | $0.86 \pm 0.00$ | $0.01 \pm 0.01$ | $0.06 \pm 0.01$ | $0.13 \pm 0.02$ | $05.96 \pm 01.16$ |
| MOLER-2000 | $0.55 \pm 0.01$ | $0.86 \pm 0.00$ | $0.86 \pm 0.00$ | $0.02 \pm 0.00$ | $0.08 \pm 0.02$ | $0.11 \pm 0.00$ | $07.62 \pm 01.80$ |
| DIGRESS | $0.77 \pm 0.01$ | $0.86 \pm 0.00$ | $0.86 \pm 0.00$ | $0.05 \pm 0.00$ | $0.09 \pm 0.01$ | $0.61 \pm 0.02$ | $32.51 \pm 00.20$ |
| EDM-SYCO | $0.59 \pm 0.00$ | $0.87 \pm 0.00$ | $0.87 \pm 0.00$ | $0.003 \pm 0.00$ | $0.19 \pm 0.01$ | $0.06 \pm 0.02$ | $04.49 \pm 00.22$ |

## G.4 CONDITIONAL VS UNCONDITIONAL GENERATION

In addition to the conditional generation experiment from the main text, we perform an extra experiment to evaluate our approach and compare it to the unconditional model, which does not consider the conditioning value. We consider four properties: (1) the penalized LogP score (PLogP), a compound's solubility and synthetic accessibility; (2) QED, a compound's drug-likeness; (3) DRD2, an estimate of the biological activity against dopamine type 2 receptors; (4) TPSA, a drug's ability to permeate cells.

We generate 10,000 valid molecules for each property and report the mean absolute error between the targets and the estimated values for the generated molecules measured by RDKit. The results are shown in Table 11. Our guidance mechanism achieves an average improvement factor of more than 5.5 times compared to the unconditional model across all properties.

Table 11: Mean absolute error between the targets and the property values of 10,000 generated molecules using regressor guidance, and the improvement factor compared to unconditional generation. We report standard deviation across 3 runs with different random seeds.

| | PLogP | QED | DRD2 | TPSA |
|---|---|---|---|---|
| UNCONDITIONAL | $2.2440 \pm 0.0018$ | $0.1530 \pm 0.0002$ | $0.0148 \pm 0.0000$ | $25.9017 \pm 0.0206$ |
| GUIDANCE | $0.3557 \pm 0.0042$ | $0.0350 \pm 0.0004$ | $0.0051 \pm 0.0001$ | $03.0382 \pm 0.0164$ |
| IMP. FACTOR ($\uparrow$) | $6.3087$ | $4.3714$ | $2.9020$ | $08.5253$ |

## G.5 CONDITIONAL GENERATION WITH CONDITIONAL DIFFUSION MODELS AND REGRESSOR-FREE GUIDANCE

In this section, we benchmark additional approaches for conditional generation presented in Section 6.2. In the main text, we reported results with the regressor guidance algorithm on top of an unconditional diffusion model inspired by the classifier guidance algorithm (Dhariwal & Nichol, 2021). Here, we additionally test a conditional diffusion model, which is a simple extension to the unconditional model, obtained by adding the target property to the node features and training a new model (Hoogeboom et al., 2022), referred to as Conditional Diffusion in Table 12. Inspired by the classifier-free guidance algorithm (Ho & Salimans, 2022), we train another variant of the conditional model by dropping the target property 15% of the time during training. This allows us to sample from the same model in a conditional or unconditional way, and by combining the two modes, we can guide the generation process towards the target properties, similar to the regressor-guidance case. We refer to this case as Regressor-Free Guidance in Table 12. We additionally test guiding the conditional model with an external regressor model (Regressor Guidance Conditional). All results for these models are shown in Table 12. Note that we experimented with smaller models than the one reported in the previous experiment due to the computational costs associated with training and running these models. Therefore, these numbers are not directly comparable to the numbers from the previous

experiment. However, note that the best model presented here, the Regressor Guidance Conditional, has an MAE of 0.36, which is very close to the number from the previous experiment (0.3581), even though the model here is smaller. We expect that guiding the conditional model performs better if compared properly. However, it has the drawback of training a new diffusion model for each target property, compared to using the same diffusion model and one regressor that predicts all properties of interest.

Table 12: MAE between condition value $c$ and the property value computed on the generated molecules. The considered property in PLogP. $\bar{s} = \sqrt{1 - \bar{\alpha}_t}s$.

| METHOD | SCORE FUNCTION | MAE |
|---|---|---|
| UNCONDITIONAL | - | 2.24 |
| CONDITIONAL DIFFUSION | $\epsilon_\theta(z_t, t, c)$ | 0.60 |
| REGRESSOR-FREE GUIDANCE | $(s+1)\epsilon_\theta(z_t, t, c) + s\epsilon_\theta(z_t, t)$ | 0.48 |
| REGRESSOR GUIDANCE CONDITIONAL | $\epsilon_\theta(z_t, t, c) + \bar{s}\nabla_{z_t}(g_\eta(z_t, t) - c)^2$ | **0.36** |

## G.6 UNCONSTRAINED MOLECULE OPTIMIZATION

This task aims to generate molecules from scratch with as high property values as possible without any other constraints. We use this task to demonstrate the performance of the proposed guidance mechanism for property maximization, described in Equation 8. Similarly to the conditional generation task, we start from Gaussian noise and run the modified sampling algorithm to generate molecules with high properties. Table 13 shows the top-3 scores of our model's generated molecules and for all baselines from (Kong et al., 2022). Note that we achieved the best scores compared to all these baselines, even though we only generated 1,000 molecules compared to 10,000 molecules generated by these baselines. This illustrates the high performance of our method and motivates its use for the harder task of constrained optimization.

Table 13: Top-3 property values for molecules generated in the unconstrained optimization task. Results for all baselines are taken from (Kong et al., 2022). All baselines generate 10,000 molecules, while we only generate 1,000 for this task.

| METHOD | PLogP | | | QED | | |
|---|---|---|---|---|---|---|
| | 1ST | 2ND | 3RD | 1ST | 2ND | 3RD |
| JT-VAE | 5.30 | 4.93 | 4.49 | 0.925 | 0.911 | 0.910 |
| GCPN | 7.98 | 7.85 | 7.80 | **0.948** | 0.947 | 0.946 |
| MRNN | 8.63 | 6.08 | 4.73 | 0.844 | 0.796 | 0.736 |
| GRAPHAF | 12.23 | 11.29 | 11.05 | **0.948** | **0.948** | 0.947 |
| GRAPHDF | 13.70 | 13.18 | 13.17 | **0.948** | **0.948** | **0.948** |
| GA | 12.25 | 12.22 | 12.20 | 0.946 | 0.944 | 0.932 |
| MARS | 7.24 | 6.44 | 6.43 | 0.944 | 0.943 | 0.942 |
| FREED | 6.74 | 6.65 | 6.42 | 0.920 | 0.919 | 0.908 |
| H-VAE | 11.41 | 9.67 | 9.31 | 0.947 | 0.946 | 0.946 |
| F-VAE | 13.50 | 12.62 | 12.40 | **0.948** | **0.948** | 0.947 |
| PS-VAE | 13.95 | 13.83 | 13.65 | **0.948** | **0.948** | **0.948** |
| EDM-SYCO | **14.73** | **14.31** | **14.21** | **0.948** | **0.948** | **0.948** |

## G.7 FRAGMENT OCCURENCES

In this section, we test our method's ability to generate molecules containing fragments of different occurrence frequencies in the training set and compare it with several baselines. Table 14 shows the results of this experiment. Even though our model is not motif-based, it can match the distribution of fragments very well.

Table 14: Frequency of occurrence of different fragments computed on the generated molecules from different methods. Methods are split into SMILES-based, graph-based motif-level, and graph-based atom-level. The method that achieves the closest percentage to that in the training set is **highlighted**. SᴜᴄO can successfully reproduce the percentages of different fragments in its generated molecules without relying on motifs.

| Fragment | C1=CC=CC=C1 | C1CCNCC1 | C1=C[NH]N=C1 | c1ccc2c(c1)CCN2 | C1=CCCCC1 | C1CCNCC1 | C1CCCCCC1 | C1CC2CCCC(C1)C2 |
|---|---|---|---|---|---|---|---|---|
| TRAINING | 76.47% | 14.15% | 9.37% | 1.19% | 1.09% | 0.97% | 0.39% | 0.27% |
| CHARVAE | 52.98% | 4.06% | 1.09% | 0.13% | 0.40% | 2.31% | 1.21% | 0.00% |
| SM-LSTM | **75.73%** | **14.02%** | **9.73%** | **1.01%** | **1.09%** | **1.10%** | **0.34%** | **0.24%** |
| PS-VAE | 71.31% | 13.08% | 5.58% | 2.84% | **1.07%** | 2.61% | 0.27% | 0.12% |
| HɪᴇʀVAE | **75.78%** | 12.80% | 7.92% | 0.71% | 0.45% | 0.56% | 0.03% | 0.00% |
| MɪCaM | 79.47% | 11.37% | 5.74% | 0.79% | 0.94% | 2.16% | **0.37%** | 0.06% |
| JT-VAE | 80.15% | **13.84%** | **9.70%** | 1.99% | **1.07%** | 1.48% | 0.60% | 0.41% |
| MAGNᴇᴛ | 69.01% | 11.54% | 7.58% | 0.99% | 1.15% | **0.87%** | 0.23% | 0.19% |
| MoLᴇR | 78.47% | 15.12% | 7.85% | **1.26%** | 0.64% | 1.24% | 0.25% | **0.21%** |
| GCPN | 53.37% | 2.03% | 0.33% | 0.44% | 3.53% | 0.38% | 0.89% | 0.13% |
| GʀᴀᴘʜAF | 34.16% | 1.27% | 2.29% | 0.11% | 0.29% | 0.25% | 0.03% | 0.00% |
| DɪGʀᴇss | 83.96% | 11.63% | 8.03% | **1.18%** | 0.52% | **1.49%** | **0.38%** | 0.10% |
| EDM-SʏCo | **75.57%** | **12.84%** | **9.17%** | 1.02% | **0.56%** | 1.89% | 0.33% | **0.16%** |

## G.8 MODEL RUNTIME

To understand the computational cost associated with our model, we report the average runtime for training and inference in Table 15. On the ZINC250K dataset, our model is trained for 1000 epochs, which takes approximately 10 days (considering intermediate evaluations every 10 epochs), with an average of 740 seconds for training per epoch. During sampling, our model greatly benefits from running batch-wise and takes, on average, 86 seconds to generate a batch of 100 molecules, which corresponds to an amortized runtime of 0.86 seconds per single molecule. Generating molecules conditioned on target values further increases the runtime because it requires running one forward and one backward pass of the regressor model at each diffusion step, increasing the average runtime to 1.8 seconds per molecule. All numbers were computed using a single Nvidia A100 GPU.

Table 15: Average runtime of EDM-SʏCo. Numbers are on the ZINC250K dataset, the training set contains 220K molecules, and the sampling times are obtained using a batch size of 100.

| | RUNTIME |
|---|---|
| TRAINING (EPOCH) | 740 SECOND / EPOCH |
| TRAINING (TOTAL) | $\sim$ 10 DAYS |
| UNCONDITIONAL SAMPLING | 0.86 SECOND / MOLECULE |
| CONDITIONAL SAMPLING (GUIDANCE) | 1.8 SECOND / MOLECULE |

## G.9 SAMPLING SPEED VERSUS GENERATION QUALITY TRADE-OFF

Our diffusion model is trained with $T = 1000$ diffusion steps, which can be a limiting factor in some practical applications due to the resulting slow sampling process. One simple way to accelerate this process is to run a strided sampling schedule Nichol & Dhariwal (2021). The key idea is that to reduce the sampling steps from $T$ to $S$, we run the denoising network (Equation 6) every $\lceil T/S \rceil$ steps and update the noise schedule accordingly Nichol & Dhariwal (2021). In this experiment, we evaluate the quality of the molecules generated by running the model, initially trained with $T = 1000$ steps, using only 500, 250, 200, and 100 steps evenly spaced between 1 and $T$ (without any additional training). Figure 9 shows that the model performs well even with only 200 steps, which offers 5x speedup (if we neglect the cost of running the decoder, which is less than 1 step of reverse diffusion).

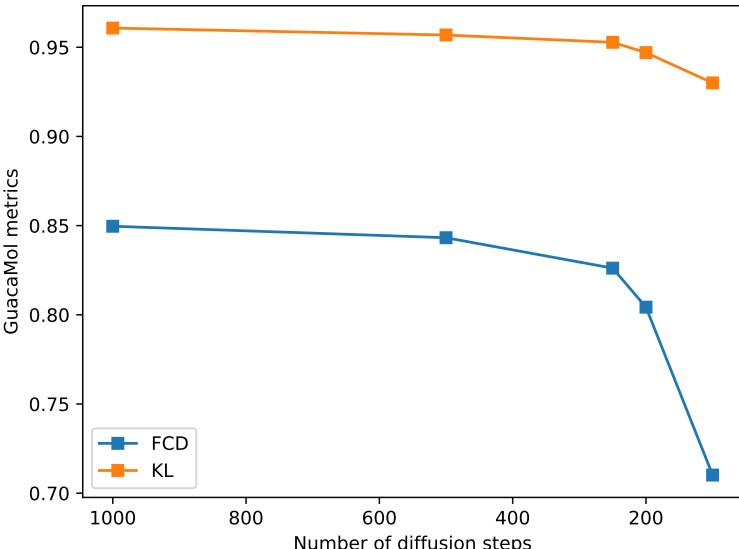

Figure 9: Effect of varying the number of reverse diffusion steps on the performance, using the same model trained for 1000 steps.

## G.10 ANALYSIS OF 3D INDUCTIVE BIAS

We analyze the inductive bias of our architecture in two steps and perform an ablation study to highlight it:

- **Chemical bias of RDKit-generated coordinates:** we train a new variant of EDM-SyCo on 3D coordinates produced by the Fruchterman-Reingold force-directed algorithm[6], which generates 3D positions solely based on the graph structure without any chemical inductive bias. We refer to it as EDM-SyCo-graph-layout.
- **Representation bias of the 3D latent embedding:** we highlight baselines that operate directly on the graphs, through discrete diffusion (DiGress) or continuous diffusion by adding Gaussian noise to the adjacency matrix (GDSS).

Results are presented in Table 16. We see that our base model performs better than EDM-SyCo-graph-layout across all metrics, which in turn outperforms GDSS and Digress on FCD and KL.

Table 16: Performance comparison of different model classes. Higher values indicate better performance for all metrics.

| Model Class | FCD (↑) | KL (↑) | Novelty (↑) | Uniqueness (↑) | Validity (↑) |
| --- | --- | --- | --- | --- | --- |
| 3D diffusion on RDKit-generated coordinates (EDM-SyCo-RDKit) | 0.85 | 0.96 | 1.00 | 1.00 | 0.88 |
| 3D diffusion on graph-based coordinates (EDM-SyCo-graph-layout) | 0.73 | 0.94 | 1.00 | 1.00 | 0.54 |
| Continuous diffusion on graph (GDSS) | 0.10 | N.A. | 1.00 | 1.00 | 0.97 |
| Discrete diffusion (DiGress) | 0.65 | 0.91 | 0.99 | 0.99 | 0.85 |

## G.11 MOVING THE EGNN FROM THE ENCODER TO THE DECODER

To understand the effect of training the diffusion model on the latent space defined by the encoder's EGNN versus on the raw synthetic coordinates, we train a new diffusion model directly on the ETKDG-generated coordinates and move the EGNN to the decoder. We note that also in this setup, training the diffusion model is necessary because, for the used datasets, there is no public ground-truth coordinates and thus no pretrained 3D diffusion models. Although both model variants achieve similar unconditional generation performance, one benefit of having an MLP-only decoder is in a

---

[6]https://networkx.org/documentation/stable/reference/generated/networkx.drawing.layout.spring_layout.html

scaffold-constrained generation setting: Given a scaffold that can be reconstructed by the autoencoder, we perform inpainting in the latent 3D space and generate a completion for this scaffold. The issue with having the EGNN in the decoder is that the bond prediction between the atoms in the scaffold will depend on the newly added atoms, and the final decoded graph is no longer guaranteed to contain the original 2D scaffold, in contrast to having an MLP-only decoder. We validate this argument by generating 1000 valid completions for a 6-Carbon-ring (smiles string: C1CCCCC1, present in 7% of the training molecules) and computing how often the decoded molecules actually contain the scaffold (Scaffolding success rate). Results are shown in Table 17.

Table 17: Performance comparison of different models on FCD, KL divergence, and Scaffolding Success Rate. Higher values indicate better performance.

| Model | FCD | KL | Scaffolding Success Rate (%) |
|---|---|---|---|
| Diffusion model on ETKDG output + EGNN-MLP-decoder | 0.84 | 0.95 | 94.2 |
| Diffusion model on EGNN output + MLP-only-decoder | 0.85 | 0.96 | 100.0 |

### G.12 MODEL SCALING WITH MOLECULE SIZE

To check the scalability of our model with respect to the molecule size compared to previous diffusion models, we compare the average runtime required to generate a single molecule using DiGress and our model. DiGress is a discrete diffusion model that operates on 2D graphs. For both models, we generate a batch of 100 molecules of size N, for an increasing N, and report the average runtime to generate one molecule. Results are shown in Table 18.

Table 18: Comparison of generation times (in seconds) for DiGress and EDM-SyCo across different molecule sizes.

| # Atoms | 10 | 20 | 30 | 40 | 50 | 60 |
|---|---|---|---|---|---|---|
| DiGress (time in seconds) | 4.05 | 12.42 | 27.04 | 46.87 | 70.73 | 114.14 |
| EDM-SyCo (time in seconds) | 0.15 | 0.49 | 1.02 | 1.75 | 2.69 | 3.66 |

### G.13 MOLECULE SIZE EXTRAPOLATION

To evaluate generalization, we generate molecules larger than every other molecule from the training set and report the validity rate in Table 19. The better performance on GuacaMol compared to ZINC250K can be explained by the higher diversity of training molecule sizes in GuacaMol. We outline some ideas that can improve the generalization capability:

- instead of modeling point clouds as fully-connected graphs for the diffusion model's EGNN, use distance cutoffs to make the number of neighbors of an atom roughly independent of the total number of atoms, as e.g. done in [3].
- modify the forward diffusion process to avoid that all atoms cluster around the CoM, e.g. by increasing the variance of the transition kernels.

Table 19: Generalization performance of models on molecules larger than the training set, with results reported as validity rates (%).

| Molecule Size | $n_{max} + 5$ | $n_{max} + 10$ | $n_{max} + 20$ |
|---|---|---|---|
| ZINC250K ($n_{mean} = 23.6$, $n_{std} = 4.4$, $n_{max} = 40$) | 45.3 | 24.5 | 21.1 |
| GuacaMol ($n_{mean} = 28.0$, $n_{std} = 7.9$, $n_{max} = 88$) | 70.1 | 55.4 | 15.9 |

