# OpenReview forum: "Lift Your Molecules: Molecular Graph Generation in Latent Euclidean Space"
_ICLR.cc/2025/Conference — ICLR 2025 Poster_

### Official Review · Reviewer_HWqN · 2024-10-27

**Soundness:** 3
**Presentation:** 3
**Contribution:** 2
**Rating:** 6
**Confidence:** 4

**Summary:**

**Summary:** The authors present SyCO as a straightforward method for generating 2D graphs using 3D generative models.

**Recommendation:** Given the notable strides yet remaining work to be clarified, discussed, or explored in this manuscript, I am currently recommending a weak reject.

**Rationale behind Recommendation:** If the authors can discuss some of the empirical benchmarking shortcomings of their model (e.g., what future steps might look like) and also include a small sample of experiments using a geometric neural network architecture other than EGNNs (to verify that the authors' ideas generalize beyond this architecture), I will consider raising my score.

**Strengths:**

- SyCO's simplicity and versatility should encourage more development in the intersection of 2D and 3D graph generation models.
- SyCO's autoencoding reconstruction performance for bond type prediction is compelling.
- The authors provide a succinct and well-motivated case study of how to add atoms to a molecule during molecular optimization via diffusion generation.
- The authors' classifier guidance is well explained and motivated for property-specific molecule generation and optimization.
- The authors' code is well documented.

**Weaknesses:**

- SyCO borrows most of its diffusion methodology from previous works [1, 2], to the best of my knowledge with adding nodes during inpainting being the only notable example to the contrary.
- Although it is simple to start with, the EDM architecture is no longer state-of-the-art for 3D molecule generation [3], suggesting the authors' results could be improved with a different choice of e.g., geometric GNN architecture. The authors seem to have a taken a wide approach to benchmarking rather than a deeper approach to seeing which geometric GNNs can best encode 2D graphs as latent 3D point clouds. This latter area of investigation could strengthen this work notably.
- The authors' conditional generation results for LogP in Table 5 and diversity throughout Table 6 seem a bit underwhelming. Besides attributing this as a flaw of molecule denoising networks, it would be helpful for the authors to expound how they think these results might be improved over time.
- The authors should discuss the pros and cons of relying on RDKit for conformer generation. For example, are there inherent biases in its distance-based coordinates estimation algorithm that might limit or impact the behavior of molecule generative models trained on its coordinate embeddings?

**References:**
1. Hoogeboom, Emiel, et al. "Equivariant diffusion for molecule generation in 3d." International conference on machine learning. PMLR, 2022.
2. Du, Yuanqi, et al. "Machine learning-aided generative molecular design." Nature Machine Intelligence (2024): 1-16.
3. Morehead, Alex, and Jianlin Cheng. "Geometry-complete diffusion for 3D molecule generation and optimization." Communications Chemistry 7.1 (2024): 150.

**Questions:**

- The authors should clarify what the error bars in Table 3 represent. Standard deviations across 3 runs of a model, as in Table 4?

---

> ### Author Response · Authors · 2024-11-21
>
> We thank the reviewer for their valuable feedback. In the following, we address the reviewer's concerns and highlight the changes to the manuscript.
>
> **Geometric GNN instead of EDM**
>
> As suggested by the reviewer, we replace EDM by the Geometry-Complete Diffusion Model (GCDM) from [1] and combine it with our framework. We used the default hyperparameters from [1] and trained for 140 epochs so far. GCDM has 6.2M parameters, so we compare it with a smaller version of EDM than the one we used in the paper, with 5.7M parameters. We find that GCDM already yields better performance than EDM for a similar model size. This shows that our framework is robust to the change of the 3D generative model, and better 3D performance corresponds to improved 2D generation performance. We add these two models to Table 1 in the manuscript.
>
> | Model |FCD ($\uparrow$)|KL ($\uparrow$)|Novelty ($\uparrow$)|Uniqueness ($\uparrow$)|Validity ($\uparrow$)|
> |-|-|-|-|-|-|
> |GCDM-SyCo|0.83|0.97|1.0|1.0|0.94|
> |EDM-SyCo|0.81|0.94|1.0|1.0|0.90|
>
> **Conditional Generation - LogP**
>
> Regarding the conditional generation results for LogP in Table 5, the results presented in the paper were obtained by using a single regressor trained to predict LogP and MW of a molecule. We now trained a model to only predict LogP and found that it improved the results. We updated the values in Table 5.
>
> | | LogP MAE |
> | -------- | -------- |
> | DiGress|0.49|
> | FreeGress|0.15|
> | EDM-SyCo|0.14|
>
> **Similarity-Constrained Optimization - Novelty**
>
> Regarding the diversity scores from Table 6, we discovered a bug in the way we computed these scores compared to the baselines. For the baseline methods, diversity was only computed for the subset of molecules that were successfully optimized (i.e. non-negative improvement for the LogP task and success for the QED and DRD2 tasks), while we have been computing the diversity over the whole set and assigning a diversity score of 0 for unsuccessful molecules, significantly underestimating the true score. We provide the previous and newly computed values and we update Table 6 in the manuscript accordingly.
> With this change, our model now achieves the best diversity score for 2 out 4 tasks, while being very close in another task. Similarly to the previous point, we expect that training different regressors for each property will further improve the optimization results and the diversity scores. We updated Table 6 accordingly.
>
> |  | LogP (sim $\geq$ 0.4) | LogP (sim $\geq$ 0.6)| QED | DRD2 |
> | - | - | - | - | - |
> | best diversity value from baselines|**0.564**|0.381|**0.477**|0.192|
> | our previous diversity values|0.555|0.360|0.163|0.083|
> | corrected diversity values|0.559|**0.388**|0.352|**0.303**|
>
> **Pros and Cons of RDKit-generated coordinates**
>
> To test the effect of the inductive bias introduced by using RDKit's coordinates, we train a new variant of EDM-SyCo on 3D coordinates produced by the Fruchterman-Reingold force-directed algorithm [2], which generates 3D positions solely based on the graph structure without any chemical inductive bias. We refer to it as EDM-SyCo-graph-layout. We find that the RDKit-based model significantly outperforms the new model, highlighting the effectiveness of this chemical inductive bias. This experiment is now in Appendix G.10. However, one drawback is that conformer generation methods would not work for non-molecular graphs such as social networks or knowledge graphs.
>
> | Model Class |FCD ($\uparrow$)|KL ($\uparrow$)|Novelty ($\uparrow$)|Uniqueness ($\uparrow$)|Validity ($\uparrow$)|
> |-|-|-|-|-|-|
> | 3D diffusion on RDKit-generated coordinates (EDM-SyCo-RDKit)|0.85|0.96|1.00|1.00|0.88|
> | 3D diffusion on graph-based coordinates (EDM-SyCo-graph-layout)|0.73|0.94|1.00|1.00|0.54|
>
> **Questions:**
> * The error bars in Table 3 represent standard deviations across 3 different training runs of differnt models, unlike in Table 4 (in table 4 we used the same model due to computational constraints on the larger GuacaMol dataset). We clarify this in the updated manuscript.
>
> **References**
> 1. Morehead, A., & Cheng, J. (2023). Geometry-complete diffusion for 3d molecule generation.
> 2. https://en.wikipedia.org/wiki/Force-directed_graph_drawing

---

> > ### Comment · Reviewer_HWqN · 2024-11-24
> > **Response to author rebuttal**
> >
> > I would like to thank the authors for their detailed response to my initial review. I believe my main concerns have been addressed. Accordingly, I have raised my score from a weak reject (5) to a weak accept (6). Overall, I think this paper explores a promising synthesis at the intersection of the development of powerful 3D geometric networks and expressive graph generative models.

---

> > > ### Author Response · Authors · 2024-11-26
> > > **Thanks**
> > >
> > > We would like to thank the reviewer again for their valuable feedback and for taking the time to consider our rebuttal and increase their score. We are happy to address any remaining questions that the reviewer might still have.

---

### Official Review · Reviewer_j8BU · 2024-11-03

**Soundness:** 3
**Presentation:** 3
**Contribution:** 3
**Rating:** 6
**Confidence:** 4

**Summary:**

This work presents a 2D molecule graph generation method that uses 3D molecule generative models by mapping 2D molecule graphs to 3D point clouds via synthetic coordinates and learning the inverse mapping using EGNN. Based on EDM, the proposed method shows superior generation quality.

**Strengths:**

- As molecules have 3D structures, using the 3D information to generate 2D molecular graphs seems intuitive and reasonable.

- The reconstruction accuracy ablation study on autoencoder shows the justification of using EGNN architecture.

- The framework can also be used for conditional generation/optimization of molecules in diverse settings, such as scaffold-constrained generation and similarity-constrained optimization.

- The experimental results show the effectiveness of this approach.

**Weaknesses:**

Although using latent Euclidean point clouds instead of a simple 2D graph structure seems to be a reasonable and effective way of modeling molecular graphs, it is likely that this approach is not scalable with respect to the molecule size. Working with latent point clouds would likely require far higher computational costs and memory compared to previous diffusion models. Especially, it seems that learning the 3D information of large molecule structures would be challenging compared to learning the 2D graph distribution. Is there any experiments or explanations on this issue?

**Questions:**

Please address the weakness mentioned above.

---

> ### Author Response · Authors · 2024-11-21
>
> We thank the reviewer for their valuable feedback. In the following, we address the reviewer's concerns and highlight the changes to the manuscript.
>
> **Scalability with respect to the molecule size**
>
> To check the scalability of our model with respect to the molecule size compared to previous diffusion models, we compare the average runtime required to generate a single molecule using DiGress [1] and our model. DiGress is a discrete diffusion model that operates on 2D graphs. For both models, we generate a batch of 100 molecules of size N, for an increasing N, and report the average runtime to generate one molecule. Results are shown in the following table. We added this experiment to Appendix G.13.
>
> | # Atoms | 10 | 20 | 30 | 40 | 50 | 60 |
> | - | - | - |  - | - | - | - |
> |DiGress (time in seconds)|4.05|12.42|27.04|46.87|70.73|114.14|
> | EDM-SyCo (time in seconds)|0.15|0.49|1.02|1.75|2.69|3.66|
>
> While the runtime for both models scales roughly quadratically with the molecule size, our model is more than one order of magnitude faster than DiGress across all tested sizes. This confirms that working with latent point clouds does not introduce additional computational or memory costs.
>
> In addition, we highlight that the second dataset we used in our experiments, GuacaMol, contains larger molecules than ZINC, with up to 88 atoms per molecule. Results in Table 4 of the manuscript show that also in this setting, our model signifcantly outperforms DiGress (0.79 vs 0.68 FCD scores).
>
> Formally, graph generative models learn a distribution of $\mathcal{O}(N^2)$ random variables due to the quadratic number of edges, where $N$ is the number of nodes in the graph. On the other hand, point cloud generative models need only to learn a distribution of $\mathcal{O}(N)$ random variables as they do not model edge distributions. In our experiments, we use GNNs to process point clouds and treat them as fully connected graphs, increasing the runtime to $\mathcal{O}(N^2)$. As we show in the previous table, this is computationally feasible for medium-sized molecules. For larger ones, the advantage of defining the diffusion process in the 3D latent space is that one could introduce distance cutoffs to define neighbors and limit the number of edges, as e.g. done in [2].
>
> Finally, in Appendix G.9, we discuss a way to accelerate the denoising process up to 5 times, while only marginally decreasing performance, by running a strided sampling schedule as proposed by [3].
>
>
> **References**
> 1. Vignac, C., Krawczuk, I., Siraudin, A., Wang, B., Cevher, V., & Frossard, P. (2022). Digress: Discrete denoising diffusion for graph generation.
> 2. Corso, G., Stärk, H., Jing, B., Barzilay, R., & Jaakkola, T. (2022). Diffdock: Diffusion steps, twists, and turns for molecular docking.
> 3. Nichol, A. Q., & Dhariwal, P. (2021, July). Improved denoising diffusion probabilistic models.

---

> ### Author Response · Authors · 2024-11-28
>
> Dear Reviewer,
>
> Thank you for raising the important concern about the scalability of our approach with respect to the molecule size. We hope to have addressed your concerns in our rebuttal and look forward to your feedback. We are happy to answer any remaining questions.
>
> Best regards,
>
> Authors

---

### Official Review · Reviewer_8xjK · 2024-11-04

**Soundness:** 3
**Presentation:** 4
**Contribution:** 3
**Rating:** 8
**Confidence:** 3

**Summary:**

This paper proposes a new method for 2D molecular graph generation using 3D molecular diffusion models, using the following steps:
- Train an autoencoder on paired (2D molecular graph, latent 3D molecular graphs).
  - The encoder consists of two parts:
       - First, 3D coordinates for all atoms are obtained using ETKDG, an well-known 3D conformation generation algorithm. No learning is required here.
       - Then, an EGNN is used to map the 3D molecular graph to a latent 3D graph, with updated node features and coordinates. The idea here is to incorporate neighborhood information for easier decoding of atom and bond types (as seen in Appendix G.1).
   - The decoder consists of two MLPs: one to predict atom types, and another to predict the bonding between the atoms.
The encoder and decoder are trained in the standard variational (VAE) framework.
* Train a diffusion model to generate latent 3D molecular graphs.
Here, the authors use the well-known EDM model (Hoogeboom, et al. 2022).
While the task is similar to that of standard 3D molecular graph generation, here the latent 3D molecular graphs do not necessarily correspond to anything physical)
* To sample:
    - Run the diffusion model, starting from noise, to generate latent 3D molecular graphs.
    - Run the decoder on the latent 3D molecular graphs to get 2D molecular graphs.

First, the authors compare their framework (EDM-SyCO) to GeoLDM, a 3D molecular graph generation algorithm combined with a simple bond prediction algorithm to create 2D molecular graphs. They show that the performance drops significantly. However, it is unclear whether the performance drop is due to the inaccurate bond prediction, or the use of standard 3D molecular graphs instead of the author’s latent 3D molecular graphs.

Next, the authors compare their model to autoregressive baselines on the well-established ZINC250K and GuacaMol datasets. They find that their model outperforms existing all-at-once 2D graph generative models. However, the performance improvement over state-of-the art autoregressive models is unclear, especially on the larger GuacaMol dataset.

Overall though, this is an interesting idea for 2D molecular graph generation. My only real concern is that the utility of the EGNN latent space is not conclusively shown; I have mentioned two experiments in the first bullet point of my Weaknesses section that I would love to see performed.

EDIT: I am happy with the author's response and additional experiments, and have increased my score in response.

**Strengths:**

- The paper is very well-written and full of details which are hard to find in most papers on this topic.
- The overall idea is definitely interesting and novel; it matches up with geometric intuition that molecules naturally live in 3D space.
- The authors have definitely improved results for non-autoregressive models on ZINC250K.
- Experiments on inpainting and conditional generation are creative, and quite comprehensive.

**Weaknesses:**

- If the motivation for having an EGNN in the encoder is to incorporate neighborhood information for predicting atom and bond types for decoding, I would have liked a comparison to a setup where the EGNN is used as a decoder (over the simple MLPs). To summarize this scheme:
    - The encoder would just output the ETKDG-generated coordinates.
    - The decoder would be an EGNN + MLP to predict atom and bond types.
    - A diffusion model could then generate (as standard) 3D molecular graphs. In fact, any pretrained diffusion model could be used, in contrast to the author’s scheme here where the diffusion model must target the latent space of the EGNN encoder.

- In Section 6.1, I would have liked to see GeoLDM paired with a more sophisticated bond-determining algorithm such as RDKit's rdDetermineBonds (an implementation of xyz2mol) or OpenBabel. For example, as shown in Table 1 of https://openreview.net/pdf?id=MIEnYtlGyv, there can be significant variation in the results from each of these bond-determining algorithms. This could help prove the utility of the EGNN latent space for the diffusion model to target.

I would be willing to increase my score if the above experiments are performed convincingly.

- Training time seems high (~10 days on a single A100 GPU) relative to other methods; this is probably related to why the authors did not tune hyperparameters on the GuacaMol dataset. I would like a confirmation from the authors on this point.

**Questions:**

- How does your model handle chirality, since the diffusion model is E(3)-equivariant? This could motivate the use of SE(3)-equivariant GNNs instead.
- Section 6.2: Please report how the normalization of the FCD and KL divergence metrics was performed for clarity.
- How is the novelty metric so high? This is very different from the 3D molecular graph generation setting (Appendix C of https://arxiv.org/abs/2203.17003) where the novelty usually reduces as training progresses. (This might be because the datasets for 3D molecular generation such as QM9 are the exact enumeration of all compounds with a certain property. However, it is surprising to see a novelty score of 100% in any case, indicating that none of the training molecules were sampled.)
- Some experiments on generalization would be very nice to see; can you train only on smaller graphs but generate much larger ones?
- Line 503 has a typo: “... factors 2.7 ad 15.6”
- Analysis of invalid molecules would be good, especially to motivate future work.
- Comparison of training and sampling time to other models would be nice, to understand the tradeoff between compute and performance better.

---

> ### Author Response · Authors · 2024-11-21
>
> We thank the reviewer for their valuable feedback. In the following, we address the reviewer's concerns and questions and highlight the changes to the manuscript.
>
> **EGNN in the decoder**
>
> We implement the reviewer’s suggestion by training a new diffusion model on ETKDG-generated coordinates and moving the EGNN to the decoder. Training the diffusion model remains necessary because the used datasets lack public ground-truth coordinates and pretrained 3D diffusion models.
>
> While both model variants achieve similar unconditional generation performance, an MLP-only decoder offers an advantage in scaffold-constrained generation. Given a scaffold that can be reconstructed by the autoencoder, we perform inpainting in latent 3D space to complete the scaffold. Using an EGNN decoder, bond predictions between scaffold atoms depend on added atoms, potentially altering the original 2D scaffold, unlike an MLP-only decoder. We validate this by generating 1000 valid completions for a 6-carbon ring (SMILES: C1CCCCC1, present in 7% of training molecules) and computing the Scaffolding Success Rate (how often the decoded molecules actually contain the scaffold), shown in the table below. We added this experiment to Appendix G.11.
>
> |Model|FCD|KL|Scaffolding success rate|
> |-|-|-|-|
> |diffusion model on ETKDG output + EGNN-MLP-decoder|0.84|0.95|94.2%
> |diffusion model on EGNN output + MLP-only-decoder|0.85|0.96|100%
>
> **GeoLDM with more bond prediction algorithms**
>
> We evaluate GeoLDM paired with the suggested bond-determining algorithms and find that the results are significantly worse than EDM-SyCo across all metrics. This suggests that having a trainable architecture such as our autoencoder is necessary to adapt 3D diffusion models to 2D molecule generation. We include this experiment in Appendix G.12.
>
> | Model | FCD | KL | Validity
> | - | - | - | - |
> | GeoLDM + Lookup Table|0.17|0.79|0.12|
> | GeoLDM + OpenBabel|0.03|0.63|0.11|
> | GeoLDM + xyz2mol|0.05|0.57|0.17|
> | EDM-SyCo |0.85|0.96|0.88|
>
> **Hyperparameter tuning on GuacaMol**
>
> We clarify this point in Section 6.2:
> > Note that due to computational constraints, we use the same hyperparameters from ZINC250K to train our model on GuacaMol
>
> ### Questions
>
> > How does your model handle chirality, since the diffusion model is E(3)-equivariant?
>
> Chirality is a property of the 3D configuration of a molecule, not of its graph structure. Therefore, as a 2D molecule generation method, our model must be invariant to chirality. To satisfy this invariance, we use E(3)-equivariant generative model. We clarify this point in Section 3.
>
> > how the normalization of the FCD and KL divergence metrics was performed
>
> We added a new section in Appendix F.8 that provides more details on the evaluation metrics following [1]
> > [...] The KL divergence $D_{KL,i}$ is computed for each descriptor between the two sets of molecules and aggregated to a final normalized score via $\frac{1}{k}\sum_{i=1}^k \exp(-D_{KL,i})$.
>
> > [...] This distance is then normalized via $\exp(-0.2\cdot FCD)$ to lie between 0 and 1.
>
> > How is the novelty metric so high?
>
> The high novelty metric arises from the larger molecular sizes in our datasets compared to QM9. The number of possible molecules grows exponentially with size, making it unlikely to sample duplicates or training molecules. As the reviewer pointed out, QM9 is the exact enumeration of organic molecules with up to 9 heavy atoms under predefined constraints [2], while molecules from ZINC250K have more than 23 atoms on average.
>
> > can you train only on smaller graphs but generate much larger ones?
>
> To evaluate generalization, we generate molecules larger than every other molecule from the training set and report the validity rate. The better performance on GuacaMol compared to ZINC250K can be explained by the higher diversity of training molecule sizes in GuacaMol. We outline some ideas that can improve the generalization capability:
> * instead of modeling point clouds as fully-connected graphs for the diffusion model's EGNN, use distance cutoffs to make the number of neighbors of an atom roughly independent of the total number of atoms, as e.g. done in [3].
> * modify the forward diffusion process to avoid that all atoms cluster around the CoM, e.g. by increasing the variance of the transition kernels.
>
> | Molecule Size | $n_{max}+5$ | $n_{max}+10$ | $n_{max}+20$ |
> | - | - | - | - |
> | ZINC250K ($n_{mean}$=23.6, $n_{std}$=4.4, $n_{max}$=40)|45.3%|24.5%|21.1%
> | GuacaMol ($n_{mean}$=28.0, $n_{std}$=7.9, $n_{max}$=88)|70.1%|55.4%|15.9%

---

> > ### Author Response · Authors · 2024-11-21
> >
> > > Analysis of invalid molecules
> >
> > We analyze the invalid molecules generated by our model and find the following common errors raised by RDKit:
> > * Kekulization errors (~55%) [https://github.com/rdkit/rdkit/wiki/FrequentlyAskedQuestions#cant-kekulize-mol]
> > * Valency errors (~33%) [https://github.com/rdkit/rdkit/wiki/FrequentlyAskedQuestions#explicit-valence-for-atom--is-greater-than-permitted]
> > * A non-ring atom is marked aromatic (~12%)
> >
> > All these errors are due to inconsistent bond type predictions for a given molecule. Because the autoencoder achieves a near-perfect reconstruction accuracy on the training and test sets, this inconsistency is likely due to a mismatch between the training distribution and the generated 3D molecules distribution. This could be improved by using more powerful 3D generative models, or one could try to jointly train the autoencoder and the latent diffusion model to adapt the decoder to the distribution of generated molecules.
> >
> > > Comparison of training and sampling time to other models
> >
> >
> > We compare the average runtime required to generate a single molecule using DiGress and our model. DiGress is a discrete diffusion model that operates on 2D graphs. For both models, we generate a batch of 100 molecules of size N, for an increasing N, and report the average runtime to generate one molecule. Results are shown in the following table. Our model is more than one order of magnitude faster than DiGress.
> >
> > | # Atoms | 10 | 20 | 30 | 40 | 50 | 60 |
> > | - | - | - |  - | - | - | - |
> > |DiGress (time in seconds)|4.05|12.42|27.04|46.87|70.73|114.14|
> > | EDM-SyCo (time in seconds)|0.15|0.49|1.02|1.75|2.69|3.66|
> >
> > **References**
> > 1. Brown, N., Fiscato, M., Segler, M. H., & Vaucher, A. C. (2019). GuacaMol: benchmarking models for de novo molecular design.
> > 2. Vignac, C., & Frossard, P. (2021). Top-n: Equivariant set and graph generation without exchangeability
> > 3. Corso, G., Stärk, H., Jing, B., Barzilay, R., & Jaakkola, T. (2022). Diffdock: Diffusion steps, twists, and turns for molecular docking.

---

> ### Comment · Reviewer_8xjK · 2024-11-25
>
> Thank you for the response! These experiments address some of my concerns, and I have raised my score accordingly.

---

> > ### Author Response · Authors · 2024-11-26
> > **Thanks**
> >
> > We would like to thank the reviewer again for their valuable feedback and for taking the time to consider our rebuttal and increase their score. We are happy to address any remaining questions that the reviewer might still have.

---

### Official Review · Reviewer_Kh9X · 2024-11-04

**Soundness:** 3
**Presentation:** 3
**Contribution:** 3
**Rating:** 6
**Confidence:** 2

**Summary:**

The paper introduces a novel 3D latent embedding scheme for 2D molecular graph generation. It presents a set of experiments which try to show that this inductive bias is helpful for a set of 2D molecule generation tasks.

**Strengths:**

The model and experiments are clearly described. The paper acknowledges the importance of generation tasks beyond vanilla distribution matching on the space of 2D molecular graphs. The experiments are chosen to attempt to shine light on the effectiveness of the 3D latent inductive bias hypothesis.

For guided generation, the comparable results with "Translation" methods are impressive.

**Weaknesses:**

The experiments are ambiguous on whether this inductive bias is, indeed, helpful.

It appears from table 3 that results on the ZINC250K dataset are saturated. The FCD and KL values reported in Table 4 are interesting, but presented without much interpretation. Without prior expertise it is difficult to evaluate FCD scores. Furthermore, de novo generation is not a terribly useful task in practice.

The most compelling experiment presented in the paper is "Similarity-Constrained Optimization". I could see how a 3D latent would improve performance in this area, but the data presented for this section is scant.

Overall, none of the experiments prove or disprove the inductive bias hypothesis of the architecture.

**Questions:**

What is the KL divergence references at the end of page 8? The description "a variety of physiochemical descriptors" is too vague. Sorry if I missed a more precise definition elsewhere.
Is there a reason the results for "Similarity-Constrained Optimization" are not presented in more detail in the main paper? This is by far the most real-world useful application, and the one which has the highest probability of showing signal on your hypothesis (in my opinion).

---

> ### Author Response · Authors · 2024-11-21
>
> We thank the reviewer for their valuable feedback. In the following, we address the reviewer's concerns and highlight the changes to the manuscript.
>
> **Definition of KL and FCD**
>
> We added a new section in Appendix F.8 to provide more details on the computation of the KL and FCD metrics following [1]:
>
> > **KL**: The following descriptors are computed for the generated and reference molecules: BertzCT, MolLogP, MolWt, TPSA, NumHAcceptors, NumHDonors, NumRotatableBonds, NumAliphaticRings, NumAromaticRings, and similarity to nearest neighbor with ECFP4 fingerprints. The KL divergence $D_{KL,i}$ is computed for each descriptor between the two sets of molecules and aggregated to a final normalized score via $\frac{1}{k}\sum_{i=1}^k \exp(-D_{KL,i})$.
>
> > **FCD**: The FCD is computed based on the hidden representations of molecules in ChemNet, trained for predicting biological activities, similarly to the FID usually applied to image generative models. Concretely, the means and covariances of the last hidden activations of ChemNet are computed for the reference and generated molecules and the Frechet distance between them is computed. This distance is then normalized via $\exp(-0.2\cdot FCD)$ to lie between 0 and 1.
>
> **De novo generation**
>
> While de novo generation is not a very useful task in practice, it is usually a prerequisite for more complex and more useful tasks such as similarity-constrained optimization, especially in the context of diffusion models where unconditional models are guided at inference time. We clarified this point in Section 6.2.
>
> **Inductive bias**
>
> We analyze the inductive bias of our architecture in two steps and perform a new ablation study to highlight it, which we included in Appendix G.10.
> * **Chemical bias of RDKit-generated coordinates:** we train a new variant of EDM-SyCo on 3D coordinates produced by the Fruchterman-Reingold force-directed algorithm [2], which generates 3D positions solely based on the graph structure without any chemical inductive bias. We refer to it as EDM-SyCo-graph-layout.
> * **Representation bias of the 3D latent embedding:** we highlight baselines that operate directly on the graphs, through discrete diffusion (DiGress [3]) or continuous diffusion by adding Gaussian noise to the adjacency matrix (GDSS [4]).
>
> Results are presented in the following table. We see that our base model performs better than EDM-SyCo-graph-layout across all metrics, which in turn outperforms GDSS and Digress on FCD and KL.
> We are happy to provide any additional results the reviewer wants to see regarding the inductive bias of our architecture.
>
> |Model Class|FCD ($\uparrow$)|KL ($\uparrow$)|Novelty ($\uparrow$)|Uniqueness ($\uparrow$)|Validity ($\uparrow$)|
> |-|-|-|-|-|-|
> |3D diffusion on RDKit-generated coordinates (EDM-SyCo-RDKit)|0.85|0.96|1.00|1.00|0.88|
> |3D diffusion on graph-based coordinates (EDM-SyCo-graph-layout)|0.73|0.94|1.00|1.00|0.54|
> |continuous diffusion on graph (GDSS)|0.10|N.A.|1.00|1.00|0.97|
> |discrete diffusion (DiGress)|0.65|0.91|0.99|0.99|0.85|
>
> **Similarity-Constrained Optimization**
>
> We refer the reviewer to Section F.7 as well as Figures 6 and 7 in the Appendix, where more details and visualizations on this task are provided, which were not presented in the main paper due to space limit. If the reviewer wants to see any additional concrete experiments in this context, we are happy to also include them.
>
> **References**
> 1. Brown, N., Fiscato, M., Segler, M. H., & Vaucher, A. C. (2019). GuacaMol: benchmarking models for de novo molecular design.
> 2. https://en.wikipedia.org/wiki/Force-directed_graph_drawing
> 3. Vignac, C., Krawczuk, I., Siraudin, A., Wang, B., Cevher, V., & Frossard, P. (2022). Digress: Discrete denoising diffusion for graph generation.
> 4. Jo, J., Lee, S., & Hwang, S. J. (2022, June). Score-based generative modeling of graphs via the system of stochastic differential equations.

---

> > ### Comment · Reviewer_Kh9X · 2024-11-25
> >
> > The inductive bias ablations are interesting, and a good addition to the paper. They partially address my concern with showing value in the inductive bias. They are not conclusive, however, so I will keep my rating the same.

---

> > > ### Author Response · Authors · 2024-11-26
> > >
> > > We thank the reviewer again for their valuable feedback and for engaging with our rebuttal. We are happy to address any remaining questions or include additional experiments that could better highlight the inductive bias of our method.

---

### Meta-Review · Area_Chair_oYpc · 2024-12-14

**Metareview:**

**Summary**

This paper introduces SyCO, a novel method for 2D molecular graph generation that leverages 3D latent embeddings. The approach involves mapping 2D molecular graphs to 3D point clouds using synthetic coordinates generated via the ETKDG algorithm, and then learning a latent 3D representation using EGNN. It aims to incorporate 3D inductive biases into the generation process, potentially improving performance on various molecule generation tasks.

**Strengths and Weaknesses**

The strengths of the paper include its novel and intuitive approach of using 3D latent embeddings for 2D molecular graph generation, leveraging the natural geometric properties of molecules. The method is well-presented, with clear descriptions and comprehensive experiments that demonstrate its effectiveness in improving results over existing non-autoregressive models on datasets like ZINC250K.
However, there are some small issues regarding to the experiments, such as the empirical verification of the 3D inductive bias,  scalability  for larger molecules and performance improvement.

**Final Decision**

Based on the reviewers' assessments and the strengths of the paper, I recommend the acceptance of this paper.

**Additional Comments On Reviewer Discussion:**

Most reviewers had actively participated the discussion during the rebuttal period.

---

### Decision · Program_Chairs · 2025-01-22

Accept (Poster)